# From What to Why: A Multi-Agent System for Evidence-based Chemical Reaction Condition Reasoning

**Cheng Yang[1], Jiaxuan Lu[2], Haiyuan Wan[2,3], Junchi Yu[4], Feiwei Qin[1]***
[1]Hangzhou Dianzi University, [2]Shanghai Artificial Intelligence Laboratory,
[3]Tsinghua University, [4]University of Oxford

## Abstract

The chemical reaction recommendation is to select proper reaction condition parameters for chemical reactions, which is pivotal to accelerating chemical science. With the rapid development of large language models (LLMs), there is growing interest in leveraging their reasoning and planning capabilities for reaction condition recommendation. Despite their success, existing methods rarely explain the rationale behind the recommended reaction conditions, limiting their utility in high-stakes scientific workflows. In this work, we propose ChemMAS, a multi-agent system that reframes condition prediction as an evidence-based reasoning task. ChemMAS decomposes the task into mechanistic grounding, multi-channel recall, constraint-aware agentic debate, and rationale aggregation. Each decision is backed by interpretable justifications grounded in chemical knowledge and retrieved precedents. Experiments show that ChemMAS achieves 20–35% gains over domain-specific baselines and outperforms general-purpose LLMs by 10–15% in Top-1 similarity, while offering falsifiable, human-trustable rationales, which establishes a new paradigm for explainable AI in scientific discovery.

## 1 Introduction

The progress in chemistry has long relied on the ability to design chemically valid reactions that yield scientific insights (Tu et al., 2023; Ismail et al., 2022). Central to this task is selecting proper reaction condition parameters, such as solvent, temperature, catalysts, and reagent ratios, which are pivotal to reaction success, selectivity, and scalability (Ball et al., 2025; Taylor et al., 2023). The traditional approach involves extensive human labor to explore the chemical reaction space, which cannot satisfy the growing demand for efficient and safe chemical synthesis (Lyall-Brookes et al., 2025; Ali et al., 2024; Lee et al., 2025). Recent advances in deep learning and data-driven modeling have opened up new opportunities for reaction recommendation, enabling automated exploration of reaction space and the discovery of novel, scalable synthetic routes with minimal manual intervention (Ali et al., 2024; Liu et al., 2023). Early work typically trains relatively small-scale models, such as graph neural networks (Wu et al., 2020) and Transformers (Vaswani et al., 2017), from scratch, achieving strong performance when abundant labeled data are available (Wang et al., 2023).

With the rapid development of large language models (LLMs) (Naveed et al., 2025; Zhao et al., 2023), there has been a growing interest in leveraging their powerful reasoning and planning abilities for reaction condition recommendation (Bran et al., 2025). Current LLM-based approaches can be broadly categorized into retrieval-based (Zhang et al., 2024b; Chen et al., 2023) and reasoning-based approaches. Retrieval-based approaches search for similar reactions from external databases and transfer their conditions to the query reaction, which is usually enhanced by learned molecular embeddings or unsupervised chemical priors to improve retrieval quality (Andronov et al., 2023). In contrast, reasoning-based approaches directly prompt or fine-tune LLMs to infer suitable reaction conditions from molecular structures or textual descriptions (Qian et al., 2023; Zhou et al., 2025), and achieve improved zero-shot and few-shot generalization capabilities.

---

*Corresponding author: Feiwei Qin (E-mail:qinfeiwei@hdu.edu.cn).

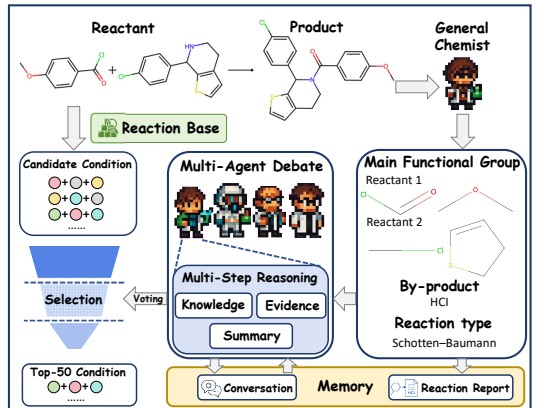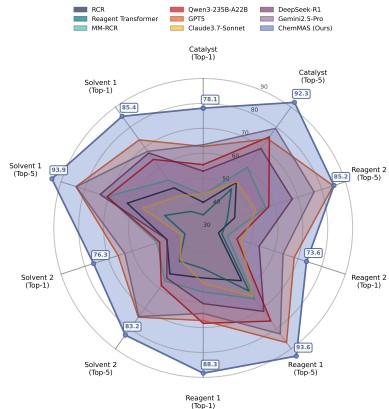

Figure 1: Overview of ChemMAS. A collaborative multi-agent system for evidence-based reaction-condition reasoning from SMILES inputs. ChemMAS demonstrates strong versatility and delivers state-of-the-art performance on reaction condition reasoning.

However, despite their success in predicting plausible reaction conditions, these approaches rarely address the deeper scientific question of why such conditions are appropriate. In the context of scientific discovery, understanding why is arguably more critical than merely predicting what. A reliable system should not only propose a solvent or temperature but also provide a mechanistic justification: Which functional group governs the reactivity? What prior experimental evidence supports this choice? Which constraints exclude alternative reagents or solvents? Without such explanatory reasoning, models risk being opaque black boxes, limiting their utility in high-stakes scientific workflows.

To tackle this challenge, we introduce ChemMAS, a multi-agent system that treats condition selection as a reasoning task grounded in chemical knowledge, mechanistic constraints, and peer deliberation. ChemMAS decomposes the problem into four collaborative stages. It first grounds chemical reactivity via mechanistic analysis, where a general chemist agent parses SMILES to identify functional groups, balance stoichiometry, and infer plausible by-products. The system then retrieves condition exemplars through multichannel queries over a structured reaction database. These candidates are refined via a tournament-style elimination process, in which agent panels conduct pairwise comparisons using memory-informed multi-step reasoning. Finally, ChemMAS aggregates rationales for each decision by combining mechanistic plausibility, retrieved evidence, and constraint checks into interpretable justifications.

By shifting from mere top-$k$ ranking to interpretable, evidence-backed reasoning, ChemMAS offers a new paradigm: one that is not only predictive but also justifiable, auditable, and suitable for closed-loop experimentation. In our evaluation, ChemMAS outperforms specialized chemical models (*e.g.*, RCR (Gao et al., 2018), Reagent Transformer (Andronov et al., 2023)) by 20-30% Top-1 similarity and surpasses leading general-purpose LLMs (*e.g.*, GPT-5, Gemini 2.5) by 10-15% on average, validating its effectiveness and robustness.

Our contributions are threefold:

- We reformulate reaction condition recommendation as evidence-based chemical reaction condition reasoning, requiring models to output not only "what"-level conditions but also "why"-level evidence.

- We introduce ChemMAS, a multi-agent system that couples chemistry-aware tool calling with multi-channel recall, multi-step mechanistic reasoning under constraint verification, and debate-based aggregation, producing interpretable, falsifiable condition reasoning.

- We benchmark ChemMAS against specialized chemical models and cutting-edge general-purpose LLMs, showing state-of-the-art performance with up to 30-point gains in Top-1 similarity and robust generalization across diverse condition types.

## 2 CHEMMAS

### 2.1 PROBLEM DEFINITION

Unlike the existing reaction condition recommendation, we formalize evidence-based reaction condition reasoning as follows. An input reaction is $\mathbf{x} = (\mathcal{R}, \mathcal{P}, \mathcal{I})$ with reactants $\mathcal{R}$, products $\mathcal{P}$, and optional context $\mathcal{I}$. A condition configuration is a structured object $\mathbf{c} \in \mathcal{C}$, where $\mathcal{C}$ may mix discrete and continuous factors. The system returns $K$ configurations $\widehat{\mathcal{C}} = \{\mathbf{c}_1, \dots, \mathbf{c}_K\}$ and a rationale for each $\rho(\mathbf{c}) = (M, S, E, \Pi)$ comprising domain reasoning $M$, verifiable checks $S$, aligned evidence $E$, and a concise derivation $\Pi$. Validity is

$$\mathsf{Valid}\big(\rho(\mathbf{c}); \mathbf{x}\big) = \mathbb{1}[\mathsf{Constr}(S) \wedge \mathsf{Align}(E; \mathbf{x}, \mathbf{c}) \geq \delta \wedge \mathsf{Coherent}(\Pi, M, E)]. \tag{1}$$

Here, $\mathsf{Constr}(S)$ is true when all hard checks in $S$ pass. $\mathsf{Align}(E; \mathbf{x}, \mathbf{c}) \in [0, 1]$ scores how well the evidence $E$ supports $(\mathbf{x}, \mathbf{c})$ using signals such as reaction-type matches, functional-group overlap, MCS alignment, or learned embeddings, with $\delta$ as a fixed threshold. $\mathsf{Coherent}(\Pi, M, E)$ verifies that the derivation $\Pi$ is logically consistent with the mechanistic summary $M$ and the evidence $E$. The indicator $\mathbb{1}$ returns 1 only when all criteria hold. The objective is

$$\max_{\widehat{\mathcal{C}}, \boldsymbol{\rho}} \sum_{\mathbf{c} \in \widehat{\mathcal{C}}} u(\mathbf{c}; \mathbf{x}) + \lambda \, \mathsf{Div}(\widehat{\mathcal{C}}) \quad \text{s.t.} \quad |\widehat{\mathcal{C}}| = K, \; \mathsf{Valid} = 1 \; \forall \mathbf{c}. \tag{2}$$

The first term accumulates a success proxy $u$ over selected configurations, where $u$ may be a calibrated yield predictor, a feasibility score, or a learned pairwise preference aggregator. The diversity term Div promotes coverage across condition dimensions to avoid mode collapse, $\lambda$ controls the trade-off between utility and diversity. The constraints enforce a fixed budget $K$ and require every selected configuration to be valid, upgrading recommendation to reasoning by demanding justified and verifiable outputs. Classical recommendation optimizes $u$ only. Our task requires each proposed $\mathbf{c}$ to carry a falsifiable, evidence-aligned certificate $\rho(\mathbf{c})$.

### 2.2 OVERVIEW

As illustrated in **Figure 2**, ChemMAS realizes the proposed reasoning framework through a multi-stage agent-based pipeline, with intermediate representations stored in a shared memory. The process begins with a General Chemist that parses the input reaction $(\mathcal{R}, \mathcal{P})$ using domain-specific tools to extract mechanistic signals, align stoichiometry, and predict reaction type. Outputs are structured into a Reaction Report written to memory. Condition hypotheses are generated via the Multi-Channel Recall module, which independently queries a historical condition database using reaction type, reactant, and product features, followed by combinatorial synthesis into candidate sets of similar conditions. The Tournament Selection phase ranks these candidates through pairwise comparisons conducted by specialized agents, each focusing on one condition dimension (*e.g.*, catalyst, solvent, reagent) under context-aware constraints. Finally, each agent engages in Multi-Step Reasoning over memory and retrieved evidence, and the Multi-Agent Debate aggregates these judgments via majority voting to produce $K$ verified configurations $\{\mathbf{c}_1, \dots, \mathbf{c}_K\}$, each paired with a rationale $\rho(\mathbf{c})$.

### 2.3 GENERAL CHEMIST

Given a chemical reaction specified by Reactant SMILES $\mathcal{R} = \{r_i\}$ and Product SMILES $\mathcal{P} = \{p_j\}$, the General Chemist ($\mathcal{A}_{Gen}$) extracts mechanistically informative priors for downstream condition prediction. The *General Chemist* agent orchestrates three tools, including *Functional Group Tagger*, *Constraint Engine*, and *Chemical Knowledge Base*, to (i) identify main functional groups, (ii) infer balanced stoichiometry and by-products, and (iii) retrieve reaction-type evidence. All outputs are written to *Memory*.

**Functional Group Tagger.** A curated library $\mathbb{L} = \{(\mathrm{name}_k, \mathrm{SMARTS}_k)\}$ of common organic motifs (*e.g.*, acyl chlorides, amines, alcohols, heteroaromatics) is used to match each $r_i$ via SMARTS substructure search, yielding $\mathcal{F}(r_i)$. The union $\mathcal{F}_{\mathcal{R}} = \bigcup_i \mathcal{F}(r_i)$ is then ranked by role salience considering electrophile/nucleophile tags, activation levels, and motif frequency across reactants. The top-ranked entries are designated as the Main FG set and stored in Memory with atom indices for downstream reference.

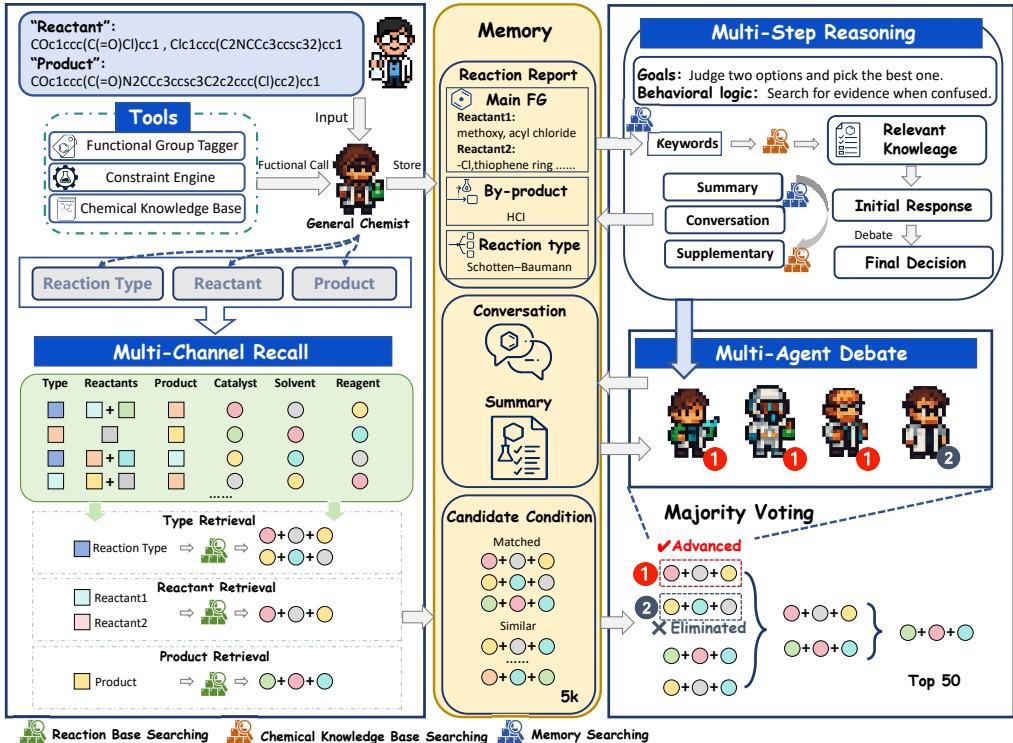

Figure 2: Architecture of ChemMAS. The left side shows how the General Chemist processes SMILES and Multi-Channel Recall retrieves reaction conditions from the Reaction Base. On the right, candidate conditions are paired and evaluated through Multi-Agent Debate, where four agents with Multi-Step Reasoning select the top-50 conditions via Tournament Selection.

**Constraint Engine.** Reactant and product molecular graphs are canonicalized (including implicit hydrogens), aligned by maximum common substructure to derive an atom mapping. An integer linear program computes stoichiometric coefficients $\boldsymbol{\nu} = (\nu_{\mathcal{R}}, \nu_{\mathcal{P}}, \nu_{\text{aux}})$. Changes on mapped atoms, combined with heuristic leaving-group rules, are used to enumerate neutral species $\mathcal{B}$, from which the most parsimonious by-product hypothesis is selected. Both the balanced equation and consistency diagnostics are written to Memory.

**Chemical Knowledge Base.** Query templates built from $\mathcal{F}_{\mathcal{R}}$, product scaffolds, and molecular identifiers are used to retrieve supporting evidence from public repositories (*e.g.*, PubChem) and a locally indexed mirror. Retrieved exemplars and co-occurrence statistics yield signal features $\mathbf{s}_{\text{ckb}} = \{s_{\text{type}}, s_{\text{role}}, s_{\text{byprod}}\}$, which support reaction type classification and by-product confirmation. The resulting labels, along with citation metadata, are stored in Memory for use in later reasoning stages.

## 2.4 MULTI-CHANNEL RECALL

We maintain a structured Reaction Base $\mathcal{D} = \{(\tau_n, \mathbf{r}_n, \mathbf{p}_n, \mathbf{c}_n)\}_{n=1}^{N}$, where each entry contains the reaction type $\tau_n$, molecular representations of reactants $\mathbf{r}_n$ and products $\mathbf{p}_n$, and a condition triple $\mathbf{c}_n = (\text{cat}, \text{sol}, \text{reag})$. Given the current reaction context $(\hat{\tau}, \mathcal{R}, \mathcal{P})$ from Memory, we perform three parallel queries including type-, reactant-, and product-centric, to obtain candidate index sets $\mathcal{S}_t, \mathcal{S}_r, \mathcal{S}_p$ (exact type match for $\mathcal{S}_t$, top-$k$ nearest neighbors by functional-group, MCS, and embedding accuracy for $\mathcal{S}_r$ and $\mathcal{S}_p$). Without any scoring or rank fusion, an entry is admitted into *Matched Conditions* if it hits on *any* of the three tags. We define the unified retrieval result as the deduplicated union:

$$\mathcal{S}_{\text{matched}} = \text{dedup}(\mathcal{S}_t \cup \mathcal{S}_r \cup \mathcal{S}_p), \tag{3}$$

and collect $\{\mathbf{c}_n : n \in \mathcal{S}_{\text{matched}}\}$ as experience-driven condition proposals. Optional feasibility filters, *e.g.*, mass/charge balance, known by-product constraints, can be applied to screen out invalid

entries. To promote diversity, we construct *Similar Conditions* via applying controlled slot-level recombination $\Pi(\mathbf{c})$ that replaces one or two elements of $\mathbf{c}$ with high co-occurrence alternatives conditioned on $(\hat{\tau}, \mathcal{F}_{\mathcal{R}})$, while removing infeasible or near-duplicate combinations. The overall candidate pool is the truncated union:

$$\mathcal{C} = \text{truncate}_{5000}\big(\mathcal{S}_{\text{matched}} \cup \mathcal{S}_{\text{similar}}\big), \tag{4}$$

which is forwarded to downstream selection and debate.

## 2.5 Candidate Pairing and Tournament Selection

We refine the initial pool of 5,000 *Candidate Conditions* into a final *Top-50* via a tournament-style knockout that emphasizes head-to-head preference (Liu et al., 2025) under comparable context rather than brittle global scoring. Let $\mathcal{C} = \{\mathbf{c}_i\}_{i=1}^{5000}$. We apply a random permutation $\pi$ and form disjoint pairs $\mathcal{P}^{(0)} = \{(\mathbf{c}_{\pi(1)}, \mathbf{c}_{\pi(2)}), \ldots, (\mathbf{c}_{\pi(4999)}, \mathbf{c}_{\pi(5000)})\}$. In round $t$, each pair $(\mathbf{a}, \mathbf{b}) \in \mathcal{P}^{(t)}$ is adjudicated by an agent panel, and the winner is determined by majority vote:

$$\text{win}(\mathbf{a}, \mathbf{b}) = \arg\max_{\mathbf{o} \in \{\mathbf{a}, \mathbf{b}\}} \sum_j \mathbb{1}[d_j = \mathbf{o}], \tag{5}$$

with a confidence-sum tie-break when necessary. Winners form $\mathcal{W}^{(t)} = \{\text{win}(\mathbf{a}, \mathbf{b})\}$, which is reshuffled and re-paired to yield $\mathcal{P}^{(t+1)} = \text{pair}(\text{shuffle}(\mathcal{W}^{(t)}))$. Iteration stops when $|\mathcal{W}^{(T)}| = 50$. We prefer this pairing-and-knockout protocol to global scoring since absolute scores are difficult to calibrate across heterogeneous condition sets and amplify noise in near-ties; head-to-head comparison avoids global calibration, anchors judgments in matched contexts, and affords linear-time selection with natural parallelism.

## 2.6 Multi-Agent Debate

**Multi-Step Reasoning.** For a candidate option $\mathbf{o} \in \{\mathbf{a}, \mathbf{b}\}$, each agent $\mathcal{A}_{Full}, \mathcal{A}_{Cat}, \mathcal{A}_{Sol}, \mathcal{A}_{Rea}$ executes an evidence-seeking chain. The agent parses the Memory *Reaction Report* (main functional groups, by-product, reaction type) to extract keywords $\kappa_j$, queries the Chemical Knowledge Base to obtain support $\Theta_j^{(0)}(\mathbf{o})$, and composes an initial assessment

$$\text{Init}_j(\mathbf{o}) = \text{LLM}\big(\kappa_j, \Theta_j^{(0)}(\mathbf{o}), \text{structured format}\big). \tag{6}$$

Across micro-rounds $u = 0, \ldots, U-1$, the agent refines its stance by reading peer summaries from the conversation buffer and re-querying when uncertainty is detected:

$$\text{Dec}_j^{(u+1)}(\mathbf{o}) = \Phi\Big(\text{Dec}_j^{(u)}(\mathbf{o}), \text{Peers}^{(u)}, \Theta_j^{(u+1)}(\mathbf{o})\Big), \tag{7}$$

where $\Phi(\cdot)$ integrates new citations, Constraint-Engine checks (*e.g.*, base required to capture HCl), and potential failure modes. Upon convergence or budget exhaustion, the agent outputs a *final decision* $d_j \in \{\mathbf{a}, \mathbf{b}\}$ with rationale saved to Memory.

**Majority Voting.** After each agent completes Multi-Step Reasoning for both $\mathbf{a}$ and $\mathbf{b}$, the panel engages in a structured debate: agents post final assessments and key citations to a shared Memory board, while a designated facilitator enforces turn-taking and prompts resolution of conflicts (*e.g.*, solvent polarity vs. nucleophile strength). The pairwise outcome is determined by majority voting as in

$$\text{win}(\mathbf{a}, \mathbf{b}) = \arg\max_{\mathbf{o} \in \{\mathbf{a}, \mathbf{b}\}} \sum_j \mathbb{1}[d_j = \mathbf{o}], \tag{8}$$

with confidence-sum tie-breaks if needed. The winning option advances to the next tournament round, losers are eliminated, and iterating over reshuffled winners progressively reduces the 5k candidates to the *Top-50*.

# 3 TWO-STAGE MULTI-TOOL COLLABORATIVE TRAINING FRAMEWORK

## 3.1 CHEMICAL TEACHING

We adopt a cold-start Supervised Fine-Tuning (SFT) recipe to endow the backbone LLM with initial Tool-Integrated Reasoning (TIR) (Dong et al., 2025) for chemical condition judgment. Given training pairs $(x_i, y_i)$, we apply the standard Supervised Fine-tuning objective on the backbone model $P_\theta$ with parameters $\theta$:

$$\mathcal{L}(\theta) = - \sum_{(x_i, y_i)} \log P_\theta(y_i \mid x_i), \tag{9}$$

where $x_i$ denotes the input prompt containing a reaction and paired candidate conditions, and $y_i$ is a structured target consisting of (i) $y_i^\tau$ : a step-wise chain that incorporates tool invocation logic and special tokens. (ii) $y_i^a$ : a concise Judgement section that independently critiques each response and declares the preferred option. The reasoning trajectory integrates two types of tools, namely *Chemical Knowledge Base searching* and *Memory searching*, serialized in special formats (e.g., `<search>`...`</search>`, `<memory>`...`</memory>`), enabling the model to learn the fundamental rules of tool invocation during the SFT process. Ultimately, this process yields a cold-start LLM $\hat{\pi}_\theta$ that learns when and how to invoke chemical tools, thereby establishing an initial capability for TIR in chemistry.

## 3.2 TOOL INCENTIVIZATION

After obtaining the cold-start model $\hat{\pi}_\theta$ via SFT, we apply tool incentivization RL to align the policy with both answer correctness and collaborative tool usage, obtaining $\pi_\theta^{\mathrm{RL}}$.

**Hierarchical Reward.** Given a valid format, we augment task accuracy Acc with a multi-tool bonus $r_M$ when both tools appear (Dong et al., 2025), otherwise we down-weight:

$$R = \begin{cases} \max(\mathrm{Acc} + r_M, \ \mathrm{Acc}), & \text{Format ok and Acc} > 0, \\ 0, & \text{Format ok and Acc} = 0, \\ -1, & \text{Otherwise}, \end{cases} \tag{10}$$

$$r_M = \begin{cases} 0.1, & \exists \ (\texttt{<search>} \ \& \ \texttt{<memory>}), \\ 0, & \text{otherwise}. \end{cases}$$

This explicitly rewards combined tool use without sacrificing correctness.

**Tool-Incentivization RL.** For each query $q$ and tool-augmented output $o$, we adopt Group Relative Policy Optimization (GRPO) (Shao et al., 2024) as our RL algorithm, which *estimates the baseline using a group of rollouts*. Concretely, we sample $G$ rollouts $\{o_i\}_{i=1}^G$, compute group-normalized advantages with a group baseline, and optimize

$$\mathcal{L}_{\mathrm{GRPO}}(\theta) = \mathbb{E}\left[\frac{1}{G}\sum_{i=1}^G \frac{1}{|o_i|}\sum_{t=1}^{|o_i|} \min\left(\rho_{i,t}\hat{A}_{i,t}, \ \mathrm{clip}(\rho_{i,t}, 1-\epsilon, 1+\epsilon)\hat{A}_{i,t}\right) - \beta \, \mathrm{D}_{\mathrm{KL}}\big[\hat{\pi}_\theta \parallel \hat{\pi}_{\mathrm{ref}}\big]\right], \tag{11}$$

where

$$\rho_{i,t}(\theta) = \frac{\hat{\pi}_\theta(o_{i,t} \mid q, o_{i,<t})}{\hat{\pi}_{\mathrm{old}}(o_{i,t} \mid q, o_{i,<t})}, \tag{12}$$

$\epsilon$ controls PPO clipping, $\beta$ weights the KL regularization to the fixed reference $\hat{\pi}_{\mathrm{ref}}$, and $\hat{A}_{i,t}$ denotes the advantage normalized with respect to the group baseline.

# 4 EXPERIMENTAL SETTINGS

## 4.1 TRAINING AND EVALUATION SETTING

All agents in ChemMAS are initialized from the same backbone, Qwen3-8B-Instruct, and are trained under a unified *Two-stage Multi-tool Collaborative Training Framework* that applies SFT and RL;

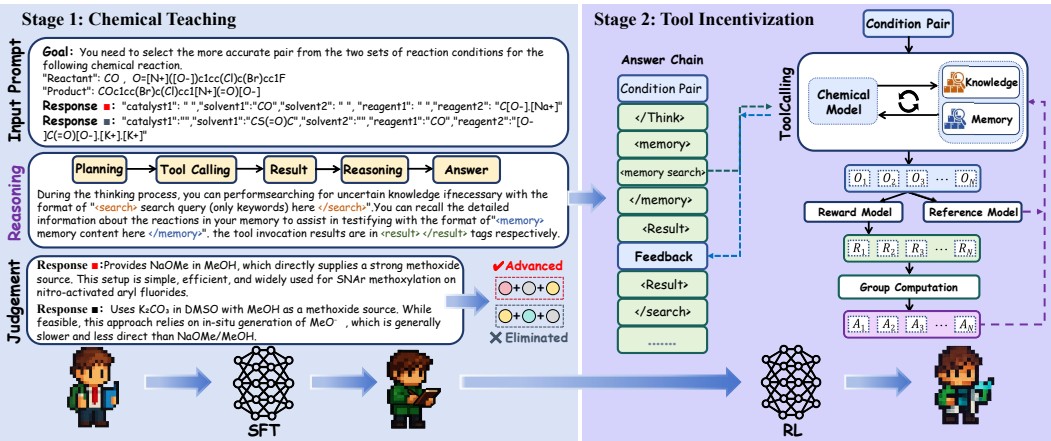

Figure 3: Two-stage Multi-tool Collaborative Training Framework of ChemMAS. Chemical Teaching uses SFT for cold-start training, enabling the LLM to master TIR, and Tool Incentivization employs RL to align the model's policy with both answer correctness and collaborative tool usage.

while the optimization protocol is shared, the learning objectives and accessible tools differ across agents. We independently trained two distinct models: one for the $\mathcal{A}_{Gen}$, and another for the multi-agent system comprising $\mathcal{A}_{Full}, \mathcal{A}_{Cat}, \mathcal{A}_{Sol}, \mathcal{A}_{Rea}$. More training details are in the Appendix.

We measure performance using Top-$k$ Similarity, defined as the maximum Tanimoto similarity between the ground truth and top-$k$ predicted candidates, averaged over a composite of molecular fingerprints (Path-based, MACCS, Morgan). This metric reflects the best structural match retrieved by the model. Details are in the Appendix.

## 4.2 DATASETS

We curate a private dataset of organic reactions, consisting of 544,591 entries represented as reaction equations in SMILES format. For each entry, the *reactants* and *products* are defined as the input, while the reaction conditions, including *catalyst1*, *solvent1*, *solvent2*, *reagent1*, and *reagent2*, are defined as the output. Based on this setting, we construct question–answer pairs and split the dataset into training, validation, and test sets with a ratio of 8:1:1.

Furthermore, we incorporated the RCR subset of ChemCoTBench (Li et al., 2025) as a lightweight public benchmark for small-scale evaluation. This subset of 90 high-quality, well-structured reaction–condition QA instances allows us to assess system generalization and stability under distribution shift. Further details on the private dataset and the ChemCoTBench-RCR subset are provided in the Appendix.

## 5 RESULTS AND DISCUSSIONS

## 5.1 MAIN RESULTS

We assessed our proposed method, ChemMAS, against a selection of current models. We compared with specialized chemical models including RCR (Gao et al., 2018), Reagent Transformer (Andronov et al., 2023), and MM RCR (Zhang et al., 2024b), which represent the latest advances in reaction-specific prediction. In addition, we benchmarked against general-purpose large language models (LLMs), such as Qwen3-235B-A22B (Yang et al., 2025), GPT5 (OpenAI), Claude 3.7 Sonnet (Anthropic, 2024), DeepSeek-R1 (Guo et al., 2025), and Gemini2.5-Pro (Comanici et al., 2025), which epitomize the cutting edge in general reasoning and knowledge transfer.

As shown in Table 2, ChemMAS surpasses both specialized chemical models and state-of-the-art LLMs across all reaction types and Top-$k$ settings. It achieves relative Top-1 similarity improvements ranging from 70% to over 90% when compared to domain-specific baselines such as RCR, Reagent Transformer, and MM RCR. Even against top-tier general-purpose LLMs like GPT-5 and

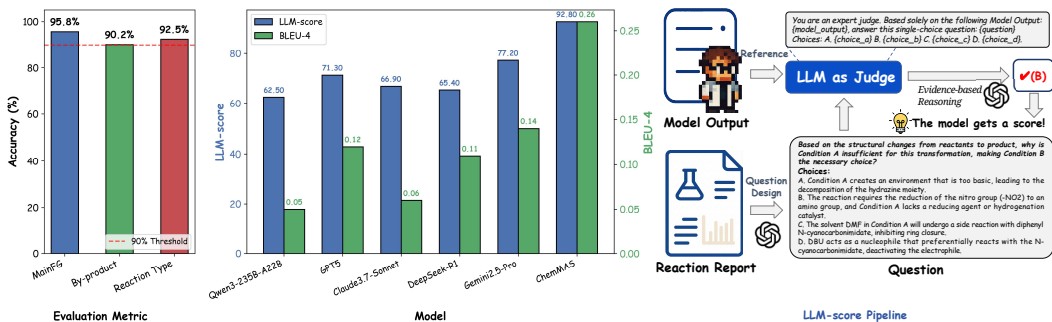

Figure 4: Model Interpretability Evaluation and Scoring Methodology. (Left) Accuracy of Chem-MAS outputs compared to human expert annotations. (Center) Human alignment performance comparison; blue bars indicate LLM-Scores and green bars indicate BLEU-4 scores. (Right) Schematic representation of the LLM-Score pipeline and the question-answering based evaluation workflow.

Gemini 2.5-Pro, ChemMAS yields consistent relative gains of 15–25% in Top-1 similarity, underscoring its strength in fine-grained mechanistic reasoning.

## 5.2 EVALUATION OF MODEL INTERPRETABILITY

To ensure the interpretability of ChemMAS, we conducted a two-level evaluation focusing on mechanistic grounding and reasoning quality. First, we validated the intermediate outputs of the General Chemist against human ground truth (Figure 4, Left). The agent demonstrates high reliability, achieving accuracies of $95.8\%$ (MainFG), $90.2\%$ (By-product), and $92.5\%$ (Reaction Type), consistently surpassing the $90\%$ threshold. This high alignment confirms that the system builds its downstream reasoning on a correct and verifiable mechanistic understanding.

Building on this foundation, we assessed the quality of the generated reasoning trajectories using a dual-metric framework comprising BLEU-4 and a semantic **LLM-Score** (Figure 4, Right). The LLM-Score employs an "LLM-as-a-Judge" mechanism to verify if the generated rationale logically supports expert-derived QA pairs. As shown in Figure 4 (Center), ChemMAS significantly outperforms general-purpose LLMs, achieving a superior LLM-Score of 92.8 compared to the 62.5–77.2 range of baselines. This substantial gap, alongside a BLEU-4 score of 0.26, demonstrates that ChemMAS offers scientifically sound explanations rather than merely plausible text.

## 5.3 GENERALIZATION EVALUATION ON OUT-OF-DISTRIBUTION DATA

To rigorously evaluate the generalization capability of our framework and assess chemical reasoning in out-of-distribution (OOD) scenarios, we conducted experiments on ChemCoT-Bench, a standardized benchmark. The primary objective of this experiment is to verify that ChemMAS does not merely rely on retrieving near-duplicate samples from the knowledge base, but truly possesses the ability to perform robust reasoning on novel reaction types.

Specifically, in the challenging Top-1 setting, as shown in Table 1, ChemMAS achieves a significant accuracy advantage. For catalyst prediction, ChemMAS attains an accuracy of 62.1%, surpassing the second-best model (Gemini 2.5-

Table 1: Generalization evaluation on ChemCoT-Bench. Top-$k$ similarity (%) for $k \in \{1, 5, 10\}$. The best and second-best results are **bolded** and underlined. Green values in parentheses show relative improvements over the second-best results.

| Model | Top-$k$ Similarity (%) | | | | | | | | |
|---|---|---|---|---|---|---|---|---|---|
| | Catalyst | | | Solvent | | | Reagent | | |
| | 1 | 5 | 10 | 1 | 5 | 10 | 1 | 5 | 10 |
| *Zero-shot LLMs* | | | | | | | | | |
| Qwen3-235B-A22B | 40.1 | 53.1 | 58.6 | 36.4 | 41.1 | 52.9 | 36.4 | 50.2 | 58.7 |
| GPT5 | 41.9 | 59.2 | 66.1 | 44.1 | 57.5 | 65.2 | 40.1 | 55.1 | 61.1 |
| Claude3.7-Sonnet | 38.5 | 56.2 | 59.1 | 40.4 | 52.1 | 61.2 | 34.3 | 48.0 | 54.3 |
| DeepSeek-R1 | 39.7 | 55.6 | 62.0 | 38.4 | 48.3 | 56.3 | 35.2 | 47.6 | 55.4 |
| Gemini2.5-Pro | 45.6 | 62.1 | 69.5 | 42.1 | 58.6 | 71.2 | 38.9 | 52.1 | 59.8 |
| **ChemMAS** | **62.1** | **68.3** | **76.1** | **57.8** | **66.5** | **76.8** | **51.2** | **59.1** | **67.7** |
| | (+16.5) | (+6.2) | (+6.6) | (+13.7) | (+7.9) | (+5.6) | (+11.1) | (+4.0) | (+6.6) |

Pro) by a margin of 16.5%. Similarly, for solvent and reagent prediction, ChemMAS outperforms the strongest competitor (GPT5) by 13.7% and 11.1%, respectively. These substantial performance gains on OOD data demonstrate that our ChemMAS framework effectively generalizes beyond the

Table 2: Main results on the private dataset. We report the Top-$k$ similarity (%) across five reaction condition types: catalyst, solvent1, solvent2, reagent1, and reagent2. Results are evaluated at $k \in \{1, 5, 10\}$. The best and second-best results are **bolded** and underlined. Green values in parentheses indicate relative improvements over the second-best results.

| Model | Top-$k$ Similarity (%) | | | | | | | | | | | | | | |
|---|---|---|---|---|---|---|---|---|---|---|---|---|---|---|---|
| | Catalyst | | | Solvent1 | | | Solvent2 | | | Reagent1 | | | Reagent2 | | |
| | 1 | 5 | 10 | 1 | 5 | 10 | 1 | 5 | 10 | 1 | 5 | 10 | 1 | 5 | 10 |
| *Pretrained Models* | | | | | | | | | | | | | | | |
| RCR | 40.3 | 52.6 | 60.7 | 49.9 | 62.1 | 68.5 | 45.3 | 52.8 | 60.3 | 50.1 | 56.2 | 63.3 | 36.4 | 43.3 | 44.9 |
| Reagent Transformer | 35.3 | 49.3 | 56.6 | 38.2 | 46.3 | 52.3 | 37.7 | 46.4 | 54.3 | 46.3 | 61.3 | 64.2 | 37.9 | 40.1 | 47.2 |
| MM RCR | 43.4 | 60.1 | 75.9 | 53.7 | 70.7 | 73.7 | 49.3 | 56.3 | 65.6 | 55.7 | 65.2 | 71.6 | 40.2 | 56.3 | 59.6 |
| *Zero-shot LLMs* | | | | | | | | | | | | | | | |
| Qwen3-235B-A22B | 55.4 | 75.2 | 77.9 | 64.0 | 70.6 | 73.7 | 48.4 | 58.6 | 64.2 | 68.3 | 76.2 | 82.7 | 44.2 | 57.7 | 60.2 |
| GPT5 | 62.7 | 74.2 | 83.2 | 73.7 | 83.7 | 86.2 | 65.9 | 74.3 | 83.6 | 67.2 | 86.9 | 90.1 | 68.4 | 84.9 | 86.1 |
| Claude3.7-Sonnet | 43.6 | 52.9 | 60.1 | 46.0 | 55.7 | 58.7 | 39.2 | 45.7 | 53.9 | 52.3 | 63.9 | 67.1 | 46.2 | 52.3 | 54.7 |
| DeepSeek-R1 | 52.8 | 69.4 | 73.2 | 67.2 | 73.5 | 78.1 | 45.2 | 54.9 | 62.2 | 60.4 | 71.4 | 75.7 | 53.6 | 67.6 | 72.3 |
| Gemini2.5-Pro | 63.4 | 79.4 | 80.5 | 68.0 | 83.6 | 86.4 | 63.1 | 74.0 | 78.6 | 64.3 | 82.6 | 90.1 | 63.7 | 76.8 | 82.2 |
| **ChemMAS** | **78.1** | **92.3** | **96.3** | **85.4** | **93.9** | **96.9** | **76.3** | **83.2** | **93.1** | **88.3** | **93.6** | **94.3** | **73.6** | **85.2** | **87.7** |
| | (+14.7) | (+12.9) | (+13.1) | (+11.7) | (+10.2) | (+10.5) | (+10.4) | (+8.9) | (+9.5) | (+20.0) | (+6.7) | (+4.2) | (+5.2) | (+0.3) | (+1.6) |

Table 3: Ablation on different components in ChemMAS. The best and second-best results are **bolded** and underlined.

| | Method | Top-$k$ Similarity (%) | | | | | | | | | | | | | | |
|---|---|---|---|---|---|---|---|---|---|---|---|---|---|---|---|---|
| | | Catalyst | | | Solvent 1 | | | Solvent 2 | | | Reagent 1 | | | Reagent 2 | | |
| | | 1 | 5 | 10 | 1 | 5 | 10 | 1 | 5 | 10 | 1 | 5 | 10 | 1 | 5 | 10 |
| Memory | w/o Main FG | 66.7 | 82.6 | 87.6 | 65.9 | 76.3 | 82.7 | 63.1 | 70.5 | 76.8 | 64.1 | 76.9 | 87.6 | 60.7 | 65.7 | 72.3 |
| | w/o By-Product | 70.3 | 88.4 | 90.1 | 78.4 | 84.1 | 89.6 | 69.7 | 76.0 | 85.9 | 74.5 | 82.8 | 90.1 | 68.2 | 74.9 | 81.6 |
| | w/o Reaction Type | 74.6 | 88.6 | 92.5 | 82.4 | 91.6 | 93.8 | 73.8 | 78.6 | 86.9 | 81.6 | 90.3 | 92.0 | 70.0 | 78.1 | 85.3 |
| Framework | w/o Multi-Agent Debate | 65.7 | 77.9 | 80.1 | 66.2 | 74.1 | 80.3 | 58.3 | 68.2 | 74.6 | 62.9 | 75.6 | 80.1 | 52.6 | 62.0 | 69.8 |
| | w/o Multi-Step Reasoning | 62.4 | 79.8 | 83.5 | 70.5 | 79.3 | 87.5 | 62.5 | 72.5 | 81.3 | 69.1 | 84.3 | 87.2 | 61.3 | 72.5 | 79.8 |
| | w/o Candidate Pairing | 74.1 | 89.7 | 92.6 | 81.6 | 90.1 | 92.5 | 72.8 | 80.4 | 89.8 | 84.2 | 89.3 | 91.5 | 71.4 | 79.4 | 82.8 |
| | **ChemMAS** | **78.1** | **92.3** | **96.3** | **85.4** | **93.9** | **96.9** | **76.3** | **83.2** | **93.1** | **88.3** | **93.6** | **94.3** | **73.6** | **85.2** | **87.7** |

training distribution, exhibiting fine-grained mechanistic reasoning rather than relying solely on memory-based retrieval.

## 5.4 ADDITIONAL QUANTITATIVE ANALYSIS

**Ablation Studies.** We conducted an ablation study to analyze the contribution of different components in ChemMAS. The ablation settings are as follows: (1) w/o Main FG, w/o By-Product, and w/o Reaction Type denote removing the corresponding elements from the Memory module; (2) w/o Multi-Agent Debate replaces multi-agent collaboration with a single-agent reasoning process, thereby eliminating conversational exchanges; (3) w/o Multi-Step Reasoning removes the iterative evidence-based reasoning chain within each agent, such that agents can only rely on prior knowledge and inter-agent debate without tool invocation; (4) w/o Candidate Pairing discards the pairwise elimination mechanism for candidate conditions, instead applying a global scoring and ranking procedure to directly select the top-50 candidates. As illustrated in **Table 3**, removing key components leads to substantial performance drops, underscoring their critical role in ChemMAS. Specifically, removing Main FG from the Memory module results in a significant decrease in performance, with an average drop of +8.4% across all reaction conditions, highlighting the crucial role of functional group extraction and analysis in reaction condition prediction. Similarly, removing Multi-Step Reasoning causes an average similarity decrease of 12.3%, underscoring the importance of evidence-based multi-round reasoning.

Table 4: Ablation study on the SFT, RL, and specific components of the hierarchical reward function, including Acc and $r_M$. The best and second-best results are **bolded** and underlined.

| Training Framework | Top-$k$ Similarity (%) | | | | | | | | | | | | | | |
|---|---|---|---|---|---|---|---|---|---|---|---|---|---|---|---|
| | **Catalyst** | | | **Solvent 1** | | | **Solvent 2** | | | **Reagent 1** | | | **Reagent 2** | | |
| | 1 | 5 | 10 | 1 | 5 | 10 | 1 | 5 | 10 | 1 | 5 | 10 | 1 | 5 | 10 |
| w/o RL | 70.6 | 88.3 | 90.4 | 82.6 | 89.4 | 90.5 | 71.2 | 80.4 | 88.5 | 84.1 | 87.5 | 90.2 | 70.2 | 82.3 | 84.5 |
| w/o SFT | 67.9 | 84.3 | 90.5 | 81.3 | 84.6 | 88.4 | 72.6 | 78.1 | 87.4 | 79.2 | 83.5 | 91.9 | 67.7 | 80.9 | 83.2 |
| w/o Acc | 72.6 | 90.8 | 93.7 | 84.1 | 91.8 | 92.1 | 76.0 | 81.6 | 91.3 | 86.7 | 90.1 | 92.0 | 72.5 | 84.1 | 86.0 |
| w/o $r_M$ | 71.9 | 89.5 | 91.2 | 83.8 | 91.5 | 91.0 | 73.5 | 81.5 | 88.7 | 84.6 | 88.6 | 90.8 | 71.6 | 82.9 | 84.9 |
| **SFT+RL** | **78.1** | **92.3** | **96.3** | **85.4** | **93.9** | **96.9** | **76.3** | **83.2** | **93.1** | **88.3** | **93.6** | **94.3** | **73.6** | **85.2** | **87.7** |

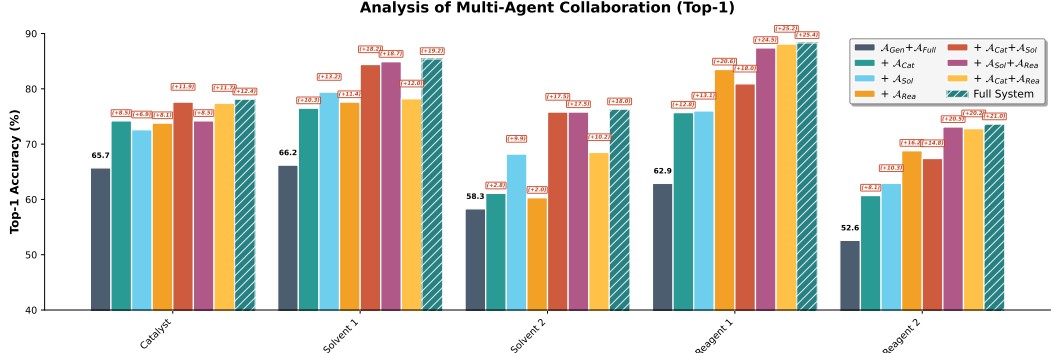

Figure 5: Multi-agent ablation: Top-1 similarity improvements across Catalyst, Solvent1/2, and Reagent1/2 when adding specialized agents on top of $\mathcal{A}_{Gen}$+$\mathcal{A}_{Full}$.

To evaluate our framework, we ablated SFT, RL, and specific hierarchical reward components. As shown in **Table 4**, removing SFT or RL significantly degrades Top-k Similarity across all conditions. Notably, excluding SFT causes a larger drop than removing RL, underscoring the importance of SFT for initialization. We further investigated the reward terms in Eq. (10) by removing task accuracy (Acc) and the multi-tool bonus ($r_M$). Results show that ablating $r_M$ impairs performance, validating the explicit reward for combined tool usage. Similarly, excluding Acc degrades results, confirming that prioritizing correctness is essential. These findings validate our two-stage framework and hierarchical reward design, where all components play complementary roles.

**Analysis of Multi-Agent Collaboration.** To assess the utility and synergy of different agents, we evaluate combinations built on the base $\mathcal{A}_{Gen}$+$\mathcal{A}_{Full}$, which are listed in **Figure 5**. Introducing specialized agents yields improvements. Specifically, $\mathcal{A}_{Cat}$ enhances performance on *Catalyst*, with an average Top-1 increase of 8.5%. $\mathcal{A}_{Sol}$ shows strong contributions on *Solvent1/2*, with an average Top-1 gain of 11.6%. $\mathcal{A}_{Rea}$ provides the largest gains on *Reagent1/2*, with an average Top-1 increase of 18.4%. When all three specialized agents are incorporated, the full system achieves macro-average Top-1 increase of 16–19% across all condition types. These results show that the specialized agents contribute substantial, domain-aligned improvements, and multi-agent debate is conducive to enhancing overall performance. For the analysis of Top-5 and Top-10, see the Appendix.

## 6 CONCLUSION

We introduce ChemMAS, a multi-agent system reframing reaction condition recommendation as evidence-based reasoning grounded in domain knowledge, mechanistic constraints, and interpretable evidence. Unlike prediction-only baselines, ChemMAS explains *why* conditions are appropriate, enhancing trust and utility. Empirically, it achieves up to 30% Top-1 similarity gains over specialized models and outperforms general LLMs. These results validate the transition from blackbox predictions to auditable decision-making. Future work will extend this framework to broader domains such as materials design and bioinformatics.

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

# Supplemental Material of ChemMAS

This document provides supplementary material to complement the main paper. It includes detailed descriptions of the ChemMAS system, prompt templates, training pipeline, additional experimental results, and reproducibility assets. Specifically:

- **Appendix A** describes how large language models (e.g., GPT-5 and Google Nano Banana) were used in writing assistance and figure generation.
- **Appendix B** summarizes related works in three areas:
    - Appendix B.1: Reaction Condition Prediction
    - Appendix B.2: LLM-Based Multi-Agent Systems
    - Appendix B.3: LLM-Based Reasoning Models
- **Appendix C** details the ChemMAS methodology, including:
    - Appendix C.1: Algorithm of ChemMAS Framework and Multi-Agent Debate
    - Appendix C.1: Two-Stage Multi-Tool Collaborative Training
- **Appendix D** outlines the experimental settings, including:
    - Appendix D.1: Training Pipeline for Agents
    - Appendix D.2: Evaluation Setting Details (Candidate Ranking)
- **Appendix E** details the evaluation protocol and metrics:
    - Appendix E.1: SMILES Canonicalization and Validity
    - Appendix E.2: Tanimoto Similarity
    - Appendix E.3: Molecular Fingerprint Types
    - Appendix E.4: Aggregate Evaluation Metrics and Top-k Similarity
- **Appendix F** introduces the dataset details:
    - Appendix F: Public Dataset (ChemCoTBench RCR subset)
    - Appendix F: Private Dataset curation and statistics
- **Appendix G** presents the prompt templates for different agents.
- **Appendix H** presents additional experimental results and discussions:
    - Appendix H.1: Additional Quantitative Results (Top-5 and Top-10 Analysis)
    - Appendix H.2: Result Visualization and qualitative analysis

## A  THE USE OF LARGE LANGUAGE MODELS

In this work, the large language model GPT-5 was used as a general-purpose tool for polishing the writing, including improving clarity and grammar. In Figure 2, the five images representing the agents and small tool icons were generated with the assistance of GPT-5[1], while the overall framework was created by the authors. The three images representing different models in Figure 3 were produced with the help of Google Nano Banana[2]. The conceptual design of both figures were entirely implemented by the authors.

## B  RELATED WORKS

### B.1  REACTION CONDITION PREDICTION

Predicting reaction conditions from reactants and products is a long-standing challenge in computer-aided synthesis. Early large-scale efforts such as (Gao et al., 2018) used feedforward neural networks trained on millions of Reaxys records to jointly predict catalysts, solvents, reagents, and temperatures, achieving promising top-k accuracies despite sparsity and label imbalance. Focusing on cross-coupling families, (Maser et al., 2021) formulated the task as multi-label ranking, developing role-specific encoders and leveraging graph-based features to yield accurate, context-aware predictions.

---

[1]https://chatgpt.com/
[2]https://www.nano-banana.ai/

To improve generalization and interpretability, (Wang et al., 2023) released benchmark datasets and proposed Parrot, a Transformer model augmented with unsupervised reaction center learning. Parrot achieved significant gains in condition similarity and temperature estimation while offering interpretable attention maps localized to reactive substructures. Separately, (Andronov et al., 2023) addressed data quality limitations by training a Molecular Transformer to impute missing reagents in USPTO reactions. Their system not only improved reagent recall but also enhanced downstream product prediction models.

Retrieval-augmented methods incorporate external knowledge to improve robustness. TextReact (Qian et al., 2023) pairs structure-based encoders with retrieved literature snippets to inform condition prediction and retrosynthesis. By integrating textual context into training, it significantly outperforms molecule-only baselines. In peptide catalysis design, (Edwards et al., 2022) proposed a semi-automated ML framework for selecting universal catalyst libraries and discovered novel, high-selectivity peptides via efficient search in a large tripeptide space. At the interface of language and chemistry, (Edwards et al., 2022) introduced MolT5, a pre-trained encoder-decoder model that translates between molecules and natural language. It supports molecule-to-caption generation and chemically constrained text-to-molecule synthesis, offering a foundation for LLM-based explainability. More recently, (Zhang et al., 2024b) proposed a text-augmented multimodal LLM framework for reaction condition recommendation. Their method jointly encodes SMILES, molecular graphs, and relevant text to achieve state-of-the-art similarity across open benchmarks and improve generalization under low-data or OOD settings. Despite these advances, current methods primarily focus on recommending what the potential reaction conditions are, but fail to provide explanatory why-level evidence for why such conditions are important or mechanistically justified.

## B.2 LLM-BASED MULTI-AGENT SYSTEMS

LLMs are increasingly deployed as autonomous agents equipped with retrieval, reasoning, and tool-use capabilities. (Boiko et al., 2023) showcased early efforts in autonomous laboratory control, with LLM agents performing iterative web search, experimental planning, and execution. (M. Bran et al., 2024) extended this direction in chemistry by coupling GPT-4 with 18 specialized tools for retrosynthesis, property prediction, and literature search. The resulting system could autonomously complete multi-step syntheses and identify new chromophores. In reaction condition recommendation, (Chen et al., 2023) leveraged retrieval-augmented generation by combining molecular similarity search, literature parsing, and in silico condition evaluation, mimicking the workflow of expert chemists.

To address hallucinations and unreliable reasoning, multi-agent collaboration has emerged as a promising direction. (Du et al., 2023) proposed a multi-agent debate framework where LLMs iteratively critique each other's answers, leading to improved factuality and robustness. (Zhu et al., 2025) benchmarked agent interactions across collaborative and competitive settings, revealing that structured debate and agent role specialization improve task success. Recent work further explores coordination protocols. (Kaesberg et al., 2025; Yang et al., 2026) found that consensus-based decision-making outperforms majority voting on complex QA tasks, while (Zhang et al., 2024a) introduced a compression pipeline that reduces inter-agent communication by up to 70% without degrading performance. (Wu et al., 2025; Cui et al., 2026; Yao et al., 2026) introduced Agentic Reasoning, a general framework for LLMs to call sub-agents (*e.g.*, web search, code execution, memory management), enabling long-horizon, tool-rich scientific workflows. Together, these systems demonstrate that combining LLMs with external tools, structured memory, and agent-level reasoning can produce scalable, verifiable pipelines for high-stakes domains. However, how to enhance the factuality and reliability of reaction condition prediction remain unexplored.

## B.3 LLM-BASED REASONING MODELS

A complementary line of work focuses on improving the reasoning capabilities of LLMs, which is essential for high-stakes decision-making and interpretability in scientific domains. In general contexts, program-aided language models (PAL) (Gao et al., 2023) execute intermediate logic through code to improve arithmetic and symbolic reasoning. CoT prompting, self-consistency, and debate-style prompting have shown broad benefits in multi-step question answering. CoMAT (Leang et al., 2024; Wang et al., 2025) proposes a mathematically annotated chain-of-thought mechanism to handle complex symbolic queries. MME-CoT (Jiang et al., 2025) benchmarks the reasoning abilities

of large multimodal models across science, math, and logic domains. In chemistry, (Tang et al., 2025) introduces a self-updating subtask library to facilitate memory-augmented chemical reasoning. It decomposes complex tasks into reusable subtasks and retrieves relevant solutions, enabling LLMs to generalize over time via experience. However, the ability to infer mechanistic or contextual rationales behind chemical reaction conditions is rarely addressed in existing works.

## C    METHOD DETAILS

### C.1    ALGORITHM OF CHEMMAS FRAMEWORK

**Multi-Agent Debate.**    In this section, we outline the overall workflow of our Multi-Agent Debate procedure. The process consists of two coordinated phases executed for each candidate pair, as illustrated in **Algorithm 1** (see also the prompt specification in **Figure 8**):

**(1) Evidence-Seeking & Refinement.** Given a pair $(\mathbf{a}, \mathbf{b})$, each agent $A_j$ initializes an evidence-seeking chain by parsing the *Reaction Report* (main functional groups, by-products, reaction type) to extract keywords, querying the Chemical Knowledge Base for citations, and composing an initial assessment. Across $U$ micro-rounds, agents iteratively refine their stance by (i) reading peer summaries from the shared buffer, (ii) re-querying the KB when uncertainty is detected, and (iii) invoking the Constraint Engine (e.g., verifying that bases are present to capture HCl). This yields a final per-agent decision $d_j \in \{\mathbf{a}, \mathbf{b}\}$ with confidence and citations.

**(2) Panel Aggregation & Tournament.** After convergence, all agents post their final assessments to the Memory board. The pairwise winner is determined by *majority voting*; ties are broken by the sum of confidences. Winners advance while losers are eliminated, and repeated rounds over reshuffled winners progressively reduce the pool to the *Top-50*. This debate-driven pipeline promotes cross-agent verification, encourages tool-grounded reasoning, and produces interpretable, citation-backed outcomes archived in Memory.

**Two-Stage Multi-Tool Collaborative Training.**    In this section, we outline the overall workflow of our Two-Stage Multi-Tool Collaborative Training pipeline. The procedure alternates two phases over multiple cycles, as illustrated in **Algorithm 2** (see also the prompt specifications in **Figure 7** and **Figure 8**):

**(1) Chemical Teaching (SFT).** Starting from the Qwen3-8B-Instruct backbone, we perform supervised fine-tuning on structured trajectories that serialize tool invocations (e.g., *search*, *memory*) before the final label. This phase teaches the model *when* and *how* to call tools and enforces a standardized output format, yielding a cold-start, tool-aware policy $\hat{\pi}_\theta$.

**(2) Tool Incentivization (RL).** Initialized from $\hat{\pi}_\theta$, we optimize the policy with GRPO using a hierarchical reward that jointly encourages (i) format validity, (ii) answer correctness, and (iii) collaborative multi-tool usage. For each query, the model samples $G$ tool-augmented rollouts; advantages are normalized with a group baseline and regularized by a KL term to a frozen reference. Policy parameters are then updated to maximize the GRPO objective.

This alternating scheme combines supervised teaching of tool protocols with reinforcement alignment for similarity and collaboration, resulting in a robust tool-aware reasoning model $\pi_\theta^{\mathrm{RL}}$ with interpretable, consistent behavior.

## D    EXPERIMENTAL SETTINGS

### D.1    TRAINING PIPELINE

For both $\mathcal{A}_{Gen}$ and the multi-agent system, we employ a two-stage optimization strategy consistent with the main framework. In the SFT stage, the AdamW optimizer is used with $\beta = (0.9, 0.95)$, an initial learning rate of $2 \times 10^{-5}$, and a weight decay of 0.1. Each model is trained for one epoch with a batch size of 128. In the subsequent RL stage, we adopt the GRPO strategy with learning rate $1 \times 10^{-6}$, KL coefficient 0.04, and number of iterations set to 1. To enhance diversity, we set the temperature parameter to 0.75 during generation. All training and inference are conducted on 8 NVIDIA A100 GPUs.

---

**Algorithm 1** Multi-Agent Debate with Multi-Step Reasoning and Majority Voting

---

**Require:** Agent set $\mathcal{A} = \{A_1, \ldots, A_m\}$; Candidates $\mathcal{C}$;
Memory: Reaction Report (main_fg, by_product, reaction_type);
Chemical Knowledge Base (KB); Constraint Engine; Micro-rounds $U$; target $K=50$.
**Output:** Top-$K$ surviving candidates

---

1: **function** MAD_TOURNAMENT($\mathcal{C}, \mathcal{A}, U, K$)
2:     **while** $|\mathcal{C}| > K$ **do**                                      ▷ pairwise tournament until Top-$K$
3:         $\mathcal{P} \leftarrow$ PAIRSHUFFLE($\mathcal{C}$)                                   ▷ form disjoint pairs
4:         $\mathcal{C}_{\text{next}} \leftarrow \emptyset$
5:         **for all** $(\mathbf{a}, \mathbf{b}) \in \mathcal{P}$ **do**
6:             $\mathcal{D} \leftarrow$ DEBATEMATCH($\mathbf{a}, \mathbf{b}, \mathcal{A}, U$)
7:             $\mathbf{o}^\star \leftarrow$ MAJORITYVOTE($\mathcal{D}$)                         ▷ winner $\mathbf{a}$ or $\mathbf{b}$
8:             $\mathcal{C}_{\text{next}} \leftarrow \mathcal{C}_{\text{next}} \cup \{\mathbf{o}^\star\}$
9:         **end for**
10:       $\mathcal{C} \leftarrow \mathcal{C}_{\text{next}}$
11:     **end while**
12:     **return** $\mathcal{C}$
13: **end function**

14: **function** DEBATEMATCH($\mathbf{a}, \mathbf{b}, \mathcal{A}, U$)
15:     $\mathcal{D} \leftarrow \emptyset$                                     ▷ per-agent final outputs and confidences
16:     **for all** $A_j \in \mathcal{A}$ **do**                           ▷ each agent reasons on both options
17:         **for all** $\mathbf{o} \in \{\mathbf{a}, \mathbf{b}\}$ **do**
18:             $\kappa_j \leftarrow$ EXTRACTKEYWORDS(Reaction Report)
19:             $\Theta_j^{(0)}(\mathbf{o}) \leftarrow$ QUERYKB($\kappa_j, \mathbf{o}$)
20:             $\text{Dec}_j^{(0)}(\mathbf{o}) \leftarrow$ COMPOSEINIT($\kappa_j, \Theta_j^{(0)}(\mathbf{o})$)
21:             **for** $u = 0$ **to** $U-1$ **do**                   ▷ micro-round refinement
22:                 $\text{Peers}^{(u)} \leftarrow$ READPEERSUMMARIES($\mathcal{A} \setminus \{A_j\}$)
23:                 **if** DETECTUNCERTAINTY($\text{Dec}_j^{(u)}(\mathbf{o}), \text{Peers}^{(u)}$) **then**
24:                     $\Theta_j^{(u+1)}(\mathbf{o}) \leftarrow$ QUERYKB($\kappa_j, \mathbf{o}$)
25:                 **else**
26:                     $\Theta_j^{(u+1)}(\mathbf{o}) \leftarrow \Theta_j^{(u)}(\mathbf{o})$
27:                 **end if**
28:                 $\Gamma_j^{(u+1)}(\mathbf{o}) \leftarrow$ CONSTRAINTCHECK($\mathbf{o}$, by_product=HCl, base-needed, $\ldots$)
29:                 $\text{Dec}_j^{(u+1)}(\mathbf{o}) \leftarrow$ UPDATEDECISION($\text{Dec}_j^{(u)}(\mathbf{o})$, $\text{Peers}^{(u)}$, $\Theta_j^{(u+1)}(\mathbf{o})$, $\Gamma_j^{(u+1)}(\mathbf{o})$)
30:             **end for**
31:         **end for**
32:         $(d_j, c_j, \text{cit}_j) \leftarrow$ FINALIZE($\text{Dec}_j^{(U)}(\mathbf{a})$, $\text{Dec}_j^{(U)}(\mathbf{b})$)
33:         WRITETOMEMORYBOARD($A_j$, $d_j$, $c_j$, $\text{cit}_j$)              ▷ store rationale/citations
34:         $\mathcal{D} \leftarrow \mathcal{D} \cup \{(A_j, d_j, c_j)\}$
35:     **end for**
36:     **return** $\mathcal{D}$
37: **end function**

38: **function** MAJORITYVOTE($\mathcal{D}$)
39:     $n_{\mathbf{a}} \leftarrow \sum_{(A_j, d_j, c_j) \in \mathcal{D}} 1\!\!\!\!1[d_j = \mathbf{a}]; \quad n_{\mathbf{b}} \leftarrow \sum_{(A_j, d_j, c_j) \in \mathcal{D}} 1\!\!\!\!1[d_j = \mathbf{b}]$
40:     **if** $n_{\mathbf{a}} \neq n_{\mathbf{b}}$ **then**
41:         **return** $\arg\max_{\mathbf{o} \in \{\mathbf{a}, \mathbf{b}\}} \{n_{\mathbf{o}}\}$
42:     **else**                                         ▷ tie-break by confidence sum
43:         $s_{\mathbf{a}} \leftarrow \sum_{(A_j, d_j, c_j) \in \mathcal{D}} c_j \cdot 1\!\!\!\!1[d_j = \mathbf{a}]$
44:         $s_{\mathbf{b}} \leftarrow \sum_{(A_j, d_j, c_j) \in \mathcal{D}} c_j \cdot 1\!\!\!\!1[d_j = \mathbf{b}]$
45:         **return** $\arg\max_{\mathbf{o} \in \{\mathbf{a}, \mathbf{b}\}} \{s_{\mathbf{o}}\}$
46:     **end if**
47: **end function**

---

**General Chemist** ($\mathcal{A}_{Gen}$). The input is limited to *Reactant* and *Product* SMILES, and the output is the predicted *Reaction Type*. During SFT, the supervision target is structured as a step-wise chain that explicitly serializes three tool invocations—*Functional Group Tagger*, *Constraint Engine*, and *Chemical Knowledge Base Searching*—before emitting the final reaction type. This design enables the model to learn *when* and *how* to call tools. In the subsequent RL stage, we apply a hierarchical reward that integrates format correctness, answer similarity, and collaborative multi-tool usage.

---

**Algorithm 2** Two-Stage Multi-Tool Collaborative Training

---

**Require:** Datasets $\mathcal{D} = \{(x_i, y_i)\}$; External tools $T$ (`<search>`, `<memory>`, ...);
        Instruction $I$; SFT epochs $E_{\text{sft}}$; RL cycles $C$; steps per cycle $S$; rollouts $G$;
        GRPO hyper-parameters ($\epsilon, \beta_{\text{KL}}$); temperature $\tau$; optimizer config.
**Output:** Trained policy $\pi_\theta^{\text{RL}}$

---

**Stage I: Chemical Teaching (SFT)**                  /* cold-start tool-aware policy */
1: Initialize backbone model $\pi_\theta \leftarrow$ Qwen3-8B-Instruct     $\triangleright$ AdamW ($\beta=(0.9, 0.95)$), lr $2\times10^{-5}$, wd 0.1,
     batch 128
2: **for** $e = 1, \ldots, E_{\text{sft}}$ **do**
3:     Sample minibatch $B \subset \mathcal{D}$
4:     Compute SFT loss $\mathcal{L}_{\text{sft}}(\theta) = -\sum_{(x,y)\in B} \log \pi_\theta(y \mid x)$     $\triangleright$ $y$ contains step-wise chain + tool tokens
     (`<search>`, `<memory>`)
5:     Update $\theta \leftarrow \theta - \eta \nabla_\theta \mathcal{L}_{\text{sft}}(\theta)$
6: **end for**
7: Freeze SFT checkpoint as reference $\hat{\pi}_{\text{ref}} \leftarrow \text{stopgrad}(\pi_\theta)$; set $\hat{\pi}_\theta \leftarrow \pi_\theta$

**Stage II: Tool Incentivization (RL with GRPO)**          /* align similarity & tool use */
1: **for** $c = 1, \ldots, C$ **do**                                   $\triangleright$ RL cycles
2:     **for** $s = 1, \ldots, S$ **do**                      $\triangleright$ optimization steps per cycle
3:        Sample a batch $D_b \subset \mathcal{D}$
4:        **for all** $q \in D_b$ **do**
5:           $q \leftarrow I \oplus q$
6:           Sample $G$ rollouts with tools at temperature $\tau$: $\{o_j\}_{j=1}^G \sim \pi_\theta(\cdot \mid q, T)$
7:           For each $o_j$, compute reward $R(o_j)$ with hierarchical scheme:
      **Format**: if invalid $\Rightarrow R(o_j) \leftarrow -1$
      **Similarity**: $\text{Acc}(o_j) \in \{0, 1\}$
      **Multi-tool bonus**: $r_M = 0.1$ if (`<search>` & `<memory>`) appear, else 0
      **Final**: if format ok, $R(o_j) = \max(\text{Acc}(o_j) + r_M, \ \text{Acc}(o_j))$
8:           Compute group-normalized advantages $\{\hat{A}_{j,t}\}$ w.r.t. group baseline
9:           Optimize GRPO objective:

$$\mathcal{L}_{\text{GRPO}}(\theta) = \frac{1}{G}\sum_{j=1}^{G}\frac{1}{|o_j|}\sum_{t=1}^{|o_j|}\min\left(\rho_{j,t}\hat{A}_{j,t}, \ \text{clip}(\rho_{j,t}, 1-\epsilon, 1+\epsilon)\hat{A}_{j,t}\right) - \beta_{\text{KL}}\,\text{D}_{\text{KL}}\left[\pi_\theta \parallel \hat{\pi}_{\text{ref}}\right]$$

10:           Update $\theta \leftarrow \theta + \eta \nabla_\theta \mathcal{L}_{\text{GRPO}}(\theta)$
11:        **end for**
12:     **end for**
13: **end for**
14: **return** $\pi_\theta^{\text{RL}}$

---

**Multi-Agent System** ($\mathcal{A}_{Full}$, $\mathcal{A}_{Cat}$, $\mathcal{A}_{Sol}$, $\mathcal{A}_{Rea}$). These role-specialized agents share the same trained backbone and are SFT on QA pairs generated in the *Candidate Pairing* stage. The supervision targets embed the invocation logic of two tools—*Chemical Knowledge Base Searching* and *Memory Searching*. The RL stage employs the same reward design to align both judgment quality and tool collaboration, ensuring that agents can deliberate effectively while remaining tool-aware.

## D.2   EVALUATION SETTING DETAILS

We evaluate general-purpose LLMs in a controlled candidate-ranking regime aligned with the Chem-MAS pipeline. Directly prompting models with only Reactant and Product SMILES yields an excessively large decision space, leading to chemically plausible yet inaccurate suggestions and a Top-1 similarity of approximately 5%. To obtain a faithful assessment, for each reaction a high-recall pool is first constructed via *Multi-Channel Recall*—aggregating reaction-base retrieval, functional-group cues, constraint heuristics, and memory lookup—to produce a Top-5000 candidate set spanning Catalyst, Solvent1, Solvent2, Reagent1, and Reagent2. Each model ranks within the same 5k pool and outputs a Top-50 list per head. All models receive identical candidate sets, instructions, and judgment interfaces, and are not permitted to modify the pool, ensuring that differences reflect discriminative ranking and evidence integration rather than retrieval coverage. This protocol mitigates search-space inflation, reduces hallucination, and provides an evaluation setting consistent with the workflow of the framework.

# E EVALUATION PROTOCOL DETAILS

In this section, we provide the formal definition of the structure-aware evaluation metrics used in our experiments. The Reaction Condition Recommendation (RCR) task requires models to predict appropriate reaction conditions given reactants and products. We evaluate the quality of predicted condition SMILES strings against ground truth annotations using molecular fingerprint similarity metrics.

## E.1 SMILES CANONICALIZATION AND VALIDITY

Prior to fingerprint calculation, all SMILES strings undergo canonicalization to ensure consistent molecular representations. Let $\hat{s}$ be the predicted SMILES and $s^*$ be the ground truth SMILES. The canonicalization procedure converts input SMILES to a standardized canonical form:

$$s_{\text{canonical}} = \text{Canonicalize}(s_{\text{input}}) \tag{13}$$

This process removes representational ambiguity. Consistent with our evaluation constraints, stereochemical information is excluded during canonicalization (`isomericSmiles=False`) to focus evaluation on constitutional structure.

The validity metric quantifies the proportion of predictions that correspond to chemically valid molecular structures:

$$\text{Validity} = \frac{1}{N} \sum_{i=1}^{N} \mathbb{1}\left[\text{IsValid}(\hat{s}_i)\right] \tag{14}$$

where $\mathbb{1}[\cdot]$ is the indicator function and $\text{IsValid}(\cdot)$ returns true if the SMILES string can be successfully parsed into a valid molecular graph by RDKit (version 2023.03 or later). Invalid predictions are assigned a similarity score of 0 for all fingerprint metrics.

## E.2 TANIMOTO SIMILARITY

All fingerprint-based similarity calculations employ the Tanimoto coefficient (Jaccard index). For two molecular fingerprints represented as bit vectors $\mathbf{A}$ and $\mathbf{B}$ (corresponding to the predicted molecule $M_p$ and ground truth molecule $M_g$), the Tanimoto similarity is defined as:

$$T(\mathbf{A}, \mathbf{B}) = \frac{|\mathbf{A} \cap \mathbf{B}|}{|\mathbf{A} \cup \mathbf{B}|} = \frac{c}{a + b - c} \tag{15}$$

where $a$ denotes the number of bits set to 1 in $\mathbf{A}$, $b$ denotes the number of bits set to 1 in $\mathbf{B}$, and $c$ denotes the number of bits set to 1 in both fingerprints simultaneously.

## E.3 MOLECULAR FINGERPRINT TYPES

We employ three complementary molecular fingerprint representations to capture different aspects of molecular structure:

**RDK Topological Fingerprint ($f_{\text{RDK}}$).** This is a path-based topological fingerprint. The algorithm enumerates all linear paths of length $l \in [1, 7]$ atoms within the molecular graph. Each path is encoded as a hash incorporating atomic numbers, bond types, and connectivity. The resulting hash values are mapped to a bit vector of length 2048. The similarity is computed as:

$$S_{\text{RDK}}(M_p, M_g) = T\left(f_{\text{RDK}}(M_p), f_{\text{RDK}}(M_g)\right) \tag{16}$$

**MACCS Keys Fingerprint ($f_{\text{MACCS}}$).** The MACCS keys fingerprint consists of 166 predefined structural keys, each corresponding to specific substructures (e.g., hydroxyl, carbonyl, aromatic rings). For each key $k_i$ ($i \in [1, 166]$), the bit is set to 1 if the substructure is present. This metric is particularly valuable for comparing molecules based on functional group composition:

$$S_{\text{MACCS}}(M_p, M_g) = T\left(f_{\text{MACCS}}(M_p), f_{\text{MACCS}}(M_g)\right) \tag{17}$$

**Morgan Circular Fingerprint ($f_{\text{Morgan}}$).** We employ Morgan fingerprints with a radius parameter of $r = 2$, equivalent to ECFP4. This algorithm captures the local chemical environment by iteratively identifying atom identifiers and their neighbors. It excels at detecting localized structural differences and functional group modifications:

$$S_{\text{Morgan}}(M_p, M_g) = T\left(f_{\text{Morgan}}(M_p, r{=}2), f_{\text{Morgan}}(M_g, r{=}2)\right) \tag{18}$$

### E.4 Aggregate Evaluation Metrics

To assess the model performance, we utilize the **Fingerprint Tanimoto Score (FTS)**, defined as the validity-weighted average of the three fingerprint similarities:

$$\text{FTS} = \left(\frac{\bar{S}_{\text{RDK}} + \bar{S}_{\text{MACCS}} + \bar{S}_{\text{Morgan}}}{3}\right) \times \text{Validity} \tag{19}$$

where $\bar{S}_{\text{type}}$ represents the mean similarity score across all test samples for that specific fingerprint type.

### E.5 Top-$k$ Similarity

Since our model generates a ranked list of candidate predictions, we also report a Top-$k$ metric. Let $\mathcal{S}(\hat{s}, s^*)$ be the pairwise similarity for a single instance, defined as the average of the three fingerprint Tanimoto coefficients (assigned as 0 if $\hat{s}$ is invalid).

For a dataset of $N$ reaction instances, where $s_i^*$ is the ground-truth SMILES and $\hat{S}_{i,k} = \{\hat{s}_{i,1}, \ldots, \hat{s}_{i,k}\}$ is the set of the top-$k$ predicted SMILES strings:

$$\text{Score@}k = \frac{1}{N}\sum_{i=1}^{N} \max_{j \in \{1,\ldots,k\}} \mathcal{S}(\hat{s}_{i,j}, s_i^*). \tag{20}$$

## F  Dataset Details

**Public Dataset.** We use the RCR subset of ChemCoTBench, which contains 90 well-structured samples covering 10 reaction types. For each reaction type, there are 9 examples: 3 focused on catalyst prediction, 3 on reagent prediction, and 3 on solvent prediction. All chemical entities (reactants/products/conditions) are represented in SMILES format to ensure consistency.

**Private Dataset.** We curate a large-scale private dataset of organic reactions to supplement existing public benchmarks and better represent real-world experimental scenarios. Sourced from the internal database of an anonymous chemical research institution, this dataset is rigorously digitized and structured, comprising 544,591 high-quality entries. Similar in nature to the USPTO-condition dataset, it encompasses a broad spectrum of known chemical reactions, reflecting a chemical space.

For data standardization, all chemical entities are represented in SMILES format. Each SMILES string is processed with RDKit to construct a molecular graph; unparseable strings are discarded. For every valid molecule, we compute the total atom count, the heavy-atom count (all non-hydrogen atoms), the molecular weight as the sum of average atomic masses (g/mol), and the exact mass as the sum of isotopic masses (g/mol). We then analyze and visualize the heavy-atom and molecular-weight distributions for the training, validation, and test sets, where the left column shows the heavy-atom counts and the right column shows the molecular weights in g/mol, as shown in Figure 6. We frame the task as a reaction condition prediction problem: for each entry, the reactants and products serve as the input, while the reaction conditions are defined as the output. To enable fine-grained prediction, the output is structured into five distinct roles: catalyst (Catalyst1), solvents (Solvent1, Solvent2), and reagents (Reagent1, Reagent2).

Based on this setting, we construct Question-Answer (QA) pairs to facilitate model training. The dataset is randomly split into training, validation, and test sets with a ratio of 8:1:1. The inclusion of this private dataset provides robust supervision signals and allows for the evaluation of model generalization in complex, realistic chemical contexts.

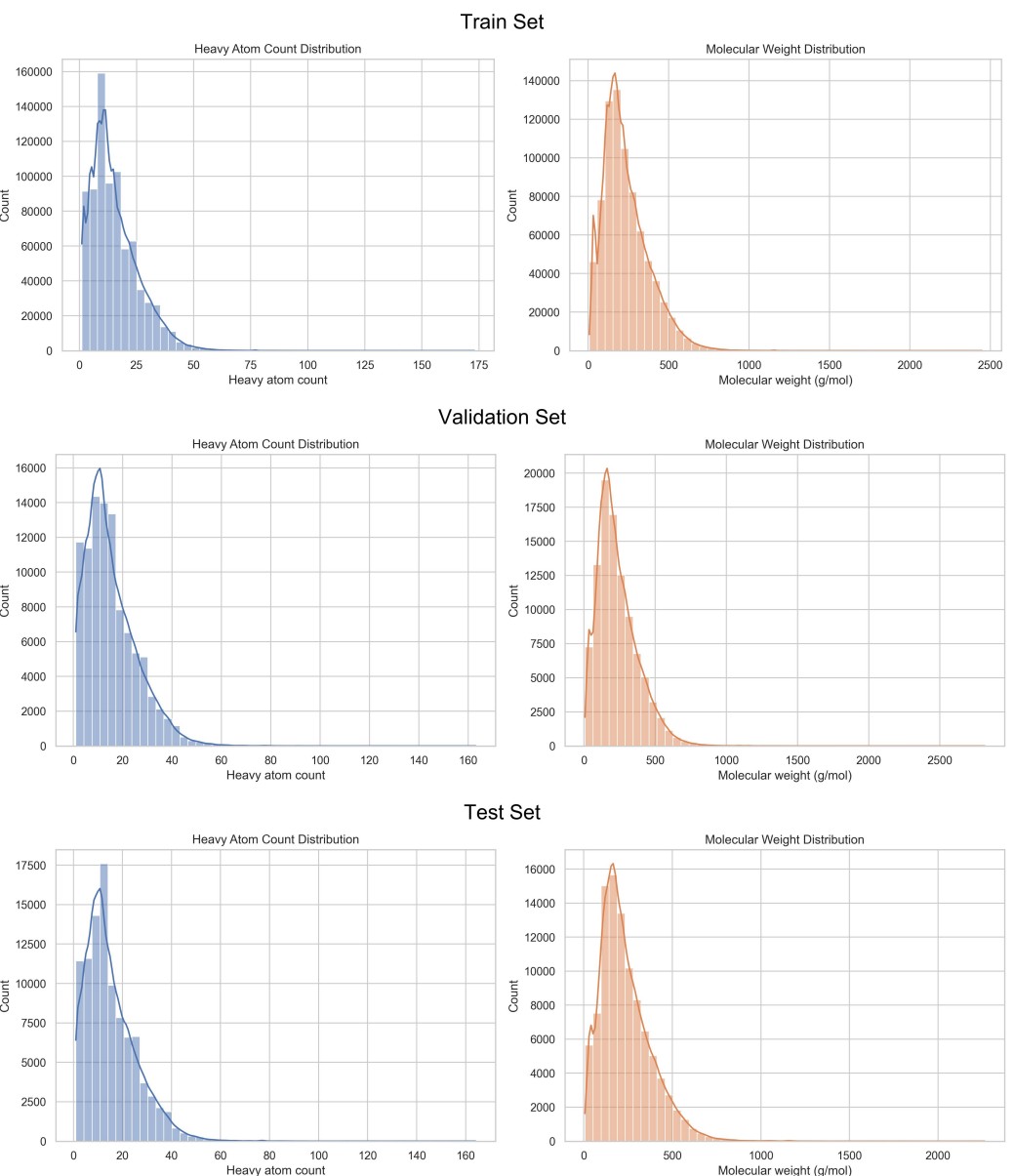

Figure 6: Heavy-atom and molecular-weight distributions for the Train, Validation, and Test sets (top to bottom). Left column: heavy atom count; right column: molecular weight (g/mol).

## G   PROMPT TEMPLATES

As shown in **Figure 7** and **Figure 8**, there are prompts for the different agents. Beyond the system-level instruction, the prompt is organized into four parts. First, the *Tool Definition* specifies the invocation schema of tools together with their expected outputs. Second, the *Interaction Protocol* describes how the agent should interleave tool calls with reasoning traces using XML-style tokens, and how the final answer must be returned in a structured format. Third, the *Task Prompt* clarifies the objectives. Finally, the *Output Format* enforces a JSON schema that standardizes the prediction into fields such as reaction type, main functional groups, by-products, and evidence. This structured prompt design enables the model to understand tool usage, maintain a consistent reasoning procedure, and produce verifiable outputs.

```
# System Prompt
You are a chemical assistant for reaction understanding and condition reasoning.
You receive Reactant and Product SMILES as inputs. You can call THREE tools:
```

```
# Tools Definition                                              General Chemist
1) Functional Group Tagger
   - Invocation: <tragger>{ "reactants":[...], "products":[...] }</tragger>
   - Purpose: extract key functional groups (FGs) from given SMILES.
   - Expected result (inside <result> ... </result>):
   {
     "reactants_fg": [...],
     "products_fg": [...],
     "main_fg": ["acyl chloride","amine", ...]
   }

2) Constraint Engine
   - Invocation: <engine>{ "reactants":[...], "products":[...] }</engine>
   - Purpose: perform atom/electron balance and infer by-products.
   - Expected result:
   {
     "balanced": true,
     "by_products": ["HCl","H2O"],
   }

3) Chemical Knowledge Base Search
   - Invocation: <search>{ "query":"..." }</search>
   - Purpose: retrieve evidence related to reaction conditions via keyword search.
   - Expected result:
   {
     "knowledge":[
       {"keyword":"amide formation","evidence":["KB:USPTO:..."]}
     ]
   }
```

```
# Interaction Protocol
- You may call tools at any time using the XML tokens above.
- Tool responses are always returned inside <result> ... </result>.
- Show reasoning process inside <think> ... </think>. For example, <think> This is the reasoning process. </think>
- Provide only the structured final answer.
```

```
# Task
Given Reactant and Product SMILES:
- Use tools to (i) extract Main Functional Groups, (ii) infer By-products, and (iii) retrieve likely Reaction Type
candidates.
- Reconcile results to choose the most plausible reaction_type.
- Output in JSON schema, wrapped inside <answer> ... </answer>.
```

```
# Output Format
<answer>
{
  "reaction_type": "string",
  "main_fg": ["string", ...],
  "by_products": ["string", ...],
  "confidence": 0.0,
  "rationale_short": "1-3 sentences summarizing the key cues.",
  "evidence": ["KB:source_or_ID", ...]
}
</answer>
```

Figure 7: Prompt for General Chemist

# H    RESULTS AND DISCUSSIONS

## H.1    ADDITIONAL QUANTITATIVE RESULTS

**Top-5 Analysis.** As shown in **Figure 9**, introducing specialized agents consistently improves Top-5 Similarity over the $\mathcal{A}_{Gen} + \mathcal{A}_{Full}$ baseline. $\mathcal{A}_{Cat}$ delivers targeted gains on *Catalyst* (+10.1%), aligning with its role specialization. $\mathcal{A}_{Sol}$ contributes the most on solvents, improving *Solvent1* and *Solvent2* by +16.4% and +13.4%, respectively. $\mathcal{A}_{Rea}$ yields the largest boosts on reagents (e.g., *Reagent1/2* with gains around +18.7% and +13.9%). When specialized agents are combined (e.g.,

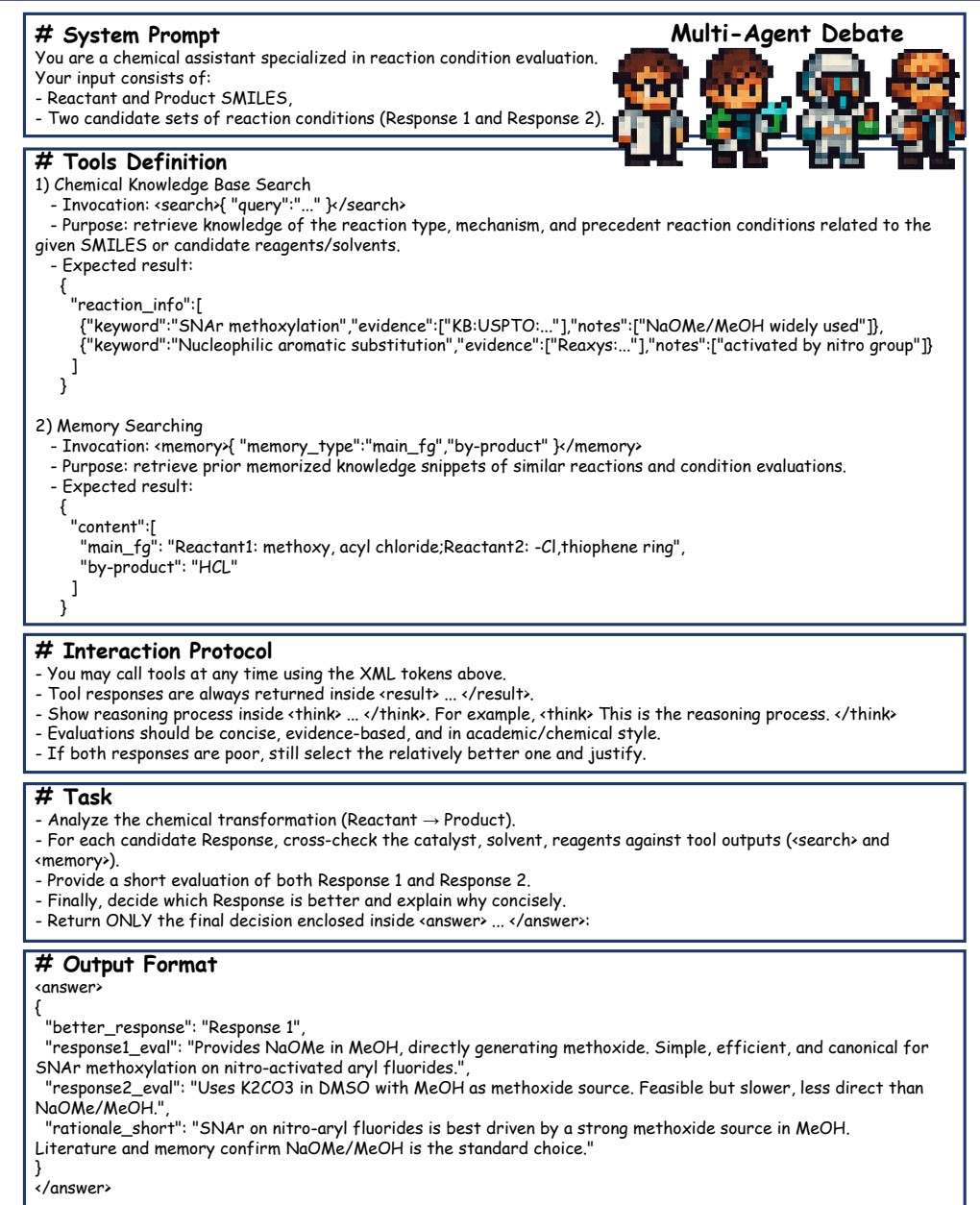

Figure 8: Prompt for Multi-Agent System

+Cat+Sol, +Sol+Rea, +Cat+Rea), the improvements remain additive and stable across condition types, and the *Full System* shows the most consistent Top-5 lift across all five heads, indicating effective collaboration among role-specialized experts.

**Top-10 Analysis.** As shown in **Figure 10**, the same trend holds for Top-10 Similarity. $\mathcal{A}_{Cat}$ most strongly benefits *Catalyst* (+13.1%). $\mathcal{A}_{Sol}$ provides clear gains on *Solvent1/2* (e.g., +10.8% and +13.6%). $\mathcal{A}_{Rea}$ again dominates on *Reagent1/2* with sizeable increments (e.g., +17.2% and +9.8%). Pairwise combinations further enhance coverage across heads, and the *Full System* achieves the highest Top-10 metrics in a macro sense, evidencing that multi-agent collaboration scales beyond single-head expertise and produces robust gains under larger candidate sets.

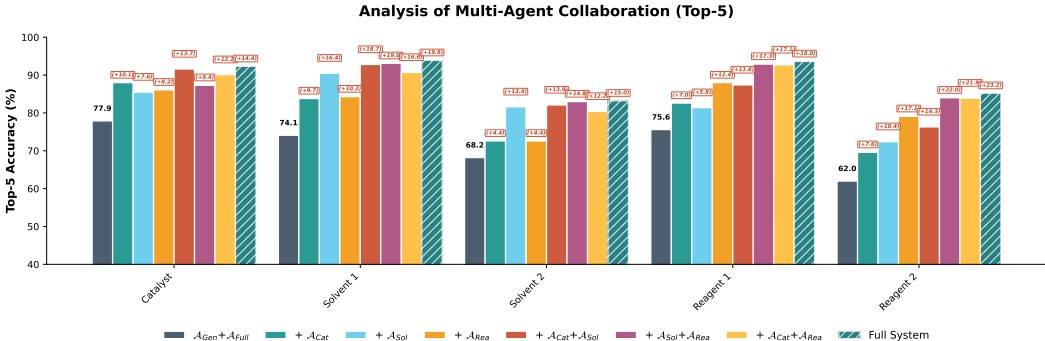

Figure 9: Multi-agent ablation: Top-5 similarity improvements across Catalyst, Solvent1/2, and Reagent1/2 when adding specialized agents on top of $\mathcal{A}_{Gen}+\mathcal{A}_{Full}$.

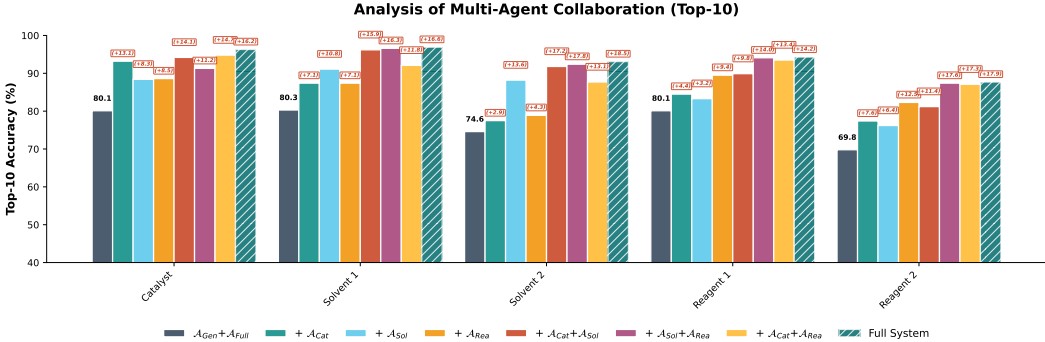

Figure 10: Multi-agent ablation: Top-10 similarity improvements across Catalyst, Solvent1/2, and Reagent1/2 when adding specialized agents on top of $\mathcal{A}_{Gen}+\mathcal{A}_{Full}$.

## H.2 RESULT VISUALIZATION

To better illustrate the performance of our framework, we visualize several representative reactions with both predicted and ground-truth conditions. As shown in **Table 5**, the predicted conditions generally align well with the ground-truth, especially for solvents and reagents that are strongly correlated with the transformation patterns in the reaction. For example, in reactions involving polar functional groups, the model consistently identifies appropriate polar solvents such as alcohols or cyclic ethers. Similarly, in palladium-catalyzed cross-coupling reactions, the model reliably predicts the use of palladium-based catalysts, demonstrating its ability to capture mechanistic priors from training data.

In cases where the predictions slightly deviate from the ground-truth, the model often proposes chemically reasonable alternatives. For instance, different bases such as potassium carbonate and cesium carbonate are interchangeable under similar conditions, and solvents like ethanol and methanol can play analogous roles. These substitutions highlight the model's flexibility in generating valid yet diverse solutions, reflecting its capacity to generalize beyond exact memorization of training examples.

Overall, the visualization confirms that the framework not only achieves high top-$k$ similarity but also produces predictions that are chemically interpretable and robust. The ability to provide both exact matches and plausible alternatives underscores the potential of our approach for assisting chemists in condition selection and experimental design.

Table 5: Visualization of several reactions with predicted (blue) vs. ground-truth (red) labels.

| Reactions | Catalyst 1 (Pred / GT) | Solvent 1 (Pred / GT) | Solvent 2 (Pred / GT) | Reagent 1 (Pred / GT) | Reagent 2 (Pred / GT) |
|---|---|---|---|---|---|
|  | | AcOH
AcOH | | Bromine
Bromine | |
|  | | Toluene
Toluene | | TEA
TEA | |
|  | | EtOH
EtOH | | Chloride
Chloride | NaOH
NaOH |
|  | Palladium
Palladium | MeOH
MeOH | | THF
THF | |
|  | | THF
THF | | AIBN
AIBN | |
|  | | MeCN
MeCN | | $K_2CO_3$
$K_2CO_3$ | |
|  | Platinum
Platinum | THF
THF | | TEA
TEA | Pyridine
Pyridine |
|  | | Toluene
Toluene | | $K_2CO_3$
$Cs_2CO_3$ | |
|  | | EtOH
MeOH | | $H_2O$
$H_2O$ | NaOEt
NaOMe |

