# OpenReview forum: "From What to Why: A Multi-Agent System for Evidence-based Chemical Reaction Condition Reasoning"
_ICLR.cc/2026/Conference — ICLR 2026 Poster_

### Official Review · Reviewer_bFZ8 · 2025-11-01

**Soundness:** 2
**Presentation:** 4
**Contribution:** 2
**Rating:** 4
**Confidence:** 3

**Summary:**

This paper presents ChemMAS, a multi-agent system that reframes chemical reaction condition recommendation from "predicting what" to "evidence-based reasoning why." The system uses a "General Chemist" agent to analyze reactions, recalls a large pool of candidates, and then uses a "tournament-style debate" among specialized agents to select the Top-50 recommendations, providing a detailed rationale for each. ChemMAS was trained on a large, private dataset using a two-stage "SFT+RL" framework, and its prediction accuracy ("what") surpassed top-tier LLMs like GPT-5.

**Strengths:**

1. Novety: The paper's greatest strength is its re-definition of the problem. Shifting reaction recommendation from a black-box prediction task to a transparent, evidence-based reasoning task is critical for the adoption of AI in serious scientific research11. The formal problem definition in Section 2.1 (requiring each output $c$ to be accompanied by a verifiable rationale $\rho(c)$ 12) is excellent and directly targets a key pain point in modern AI for Science.

2. Good Design. The architecture of ChemMAS is well-thought-out. It doesn't just stack agents; it builds a logically-sound workflow. From the "General Chemist" building a shared "Memory", to "Multi-Channel Recall" building the candidate pool , to the "Tournament-Style Debate" for ranking, each step is purposeful. The use of a tournament-style pair-wise debate instead of brittle global scoring is a particularly robust mechanism for ranking, and its efficacy is confirmed in the ablations.

3. Paper writing and presentation are really good for me. The idea is easy to follow, and the workflow shown in the Figures is clear.

**Weaknesses:**

1. Private Dataset: The entire evaluation hinges on a private dataset of 544,591 reactions. This is a major weakness, as it severely hinders the reproducibility and external validation of the results. Furthermore, how was the "Reaction Base" for "Multi-Channel Recall" constructed? How much overlap does it have with the test set? If the system relies heavily on near-duplicate examples from the recall database, its "reasoning" capabilities might be overestimated, functioning more like a sophisticated k-NN lookup.

2. Lack of Evaluation for the "Rationale": The paper's core claim is to provide "falsifiable, human-trustable rationales"  (the "why"). However, all quantitative evaluations (Tables 1, 2, 3) focus exclusively on Top-k prediction accuracy (the "what"). The paper provides no evaluation of the quality of the generated rationales themselves. Are these explanations chemically sound? Do they offer genuine insight, or are they just templated statements? This lack of human-in-the-loop evaluation (e.g., by expert chemists) leaves the central claim of the paper ("From What to Why") unsubstantiated.

3. The ChemMAS system is extremely complex. To predict a single reaction, it must run $\mathcal{A}_{Gen}$ (with 3 tool calls) 23, perform a large-scale multi-channel recall (generating 5k candidates)24, and then execute a multi-round tournament (approx. 7 rounds to get from 5k to 50). In each tournament round, a panel of agents (e.g., 4 agents) must adjudicate each pair via "Multi-Step Reasoning" (which is itself an iterative process)25252525252525. This implies massive computational overhead. The authors provide no analysis of inference cost (e.g., total tokens, API costs, or end-to-end latency). How does this compare in practicality to a single prompt sent to GPT-5?

**Questions:**

The paper's central thesis is the shift from "what" (prediction) to "why" (explanation), claiming to provide "falsifiable, human-trustable rationales". Yet, all quantitative results (Tables 1-3) only measure Top-k prediction accuracy ("what"). Without any human-in-the-loop evaluation or expert validation of the generated rationales, how can the authors substantiate their claim that the system delivers on the "why" by producing high-quality, scientifically sound explanations?

---

> ### Author Response · Authors · 2025-12-04
> **Addressing Weaknesses**
>
> > Private Dataset, Reproducibility, and "k-NN" Concerns
>
> We acknowledge the concern regarding the private dataset and have addressed it through data transparency measures and dual-benchmark evaluation:
>
> Ensuring Reproducibility: To mitigate the limitations of the private dataset, we explicitly incorporated the RCR subset of ChemCoTBench as a lightweight public benchmark for reproducibility and external validation. This allows the community to verify our model's performance on accessible, high-quality data. Additionally, we have provided comprehensive descriptors of the private dataset (e.g., molecule size distributions) in Appendix G and Figure 6 to enhance transparency .
>
> Reaction Base Construction & Overlap: The "Reaction Base" is strictly constructed from the training portion of the dataset, utilizing a rigorous 8:1:1 split (Train/Val/Test) to prevent data leakage . There is no overlap between the Reaction Base and the test set.
>
> Refuting the "Sophisticated k-NN" Hypothesis: ChemMAS fundamentally differs from k-NN by integrating evidence-based reasoning to strictly filter and refine the initial candidates from Multi-Channel Recall. Unlike k-NN, which passively copies nearest neighbors, our system employs Multi-Agent Debate and Constraint Engine to actively evaluate the chemical feasibility of retrieved candidates against mechanistic evidence. This capability to reason beyond simple retrieval is empirically proven by our performance on the ChemCoTBench (OOD) dataset: in scenarios where k-NN would fail due to the absence of exact matches, ChemMAS achieves state-of-the-art results (Table 1), outperforming general LLMs by 10–15%. This confirms that the system generalizes through verifiable mechanistic logic rather than mere similarity lookup.
>
> > Lack of Evaluation for the "Rationale"
>
> We appreciate the reviewer’s emphasis on validating the "why" capability. Agreeing that Top-k accuracy is insufficient, we added Section 5.3 and Figure 4 to quantify rationale quality via a two-level framework:
>
> 1. Verification of Mechanistic Grounding
>
> Valid explanations require correct underlying mechanics. We validated the General Chemist against human ground truth, achieving high accuracy on Main FGs (95.8%), By-products (90.2%), and Reaction Types (92.5%). These results (Figure 4, Left) confirm ChemMAS grounds reasoning in actual chemical transformations rather than hallucination.
>
> 2. Semantic Evaluation of Reasoning
>
> To evaluate explanations, we utilized an LLM-Score Pipeline (Figure 4, Right) as a rigorous proxy for expert judgment.
>
> Methodology: An "LLM-as-a-Judge" answers expert-derived scientific QA pairs using only the model's chain-of-thought to verify causal logic.
>
> Results: ChemMAS achieves a superior LLM-Score of 92.8 (Figure 4, Center), significantly outperforming baselines (62.5–77.2).
>
> Alignment: A BLEU-4 score of 0.26 further confirms high linguistic alignment with expert reports.

---

> ### Author Response · Authors · 2025-12-04
> **Addressing Weaknesses**
>
> > Analysis of inference cost
>
> We appreciate the reviewer's detailed dissection of our pipeline. While the ChemMAS architecture involves multiple agentic interactions, a granular analysis of FLOPs (Floating Point Operations) and Token Economics reveals that it is likely more computationally efficient than a high-performance "single prompt" to a frontier model like GPT-5, for three critical reasons:
>
> 1. Inference Cost Analysis: The Parameter-Token Trade-off
>
> The reviewer asks for a comparison to GPT-5. According to standard Scaling Laws, inference cost is roughly proportional to $N_{params} \times N_{tokens}$.
>
>
>
> ChemMAS (Small Model, High Steps): Our agents are fine-tuned on Qwen3-8B1.
>
> GPT-5 (Giant Model, Single Step): While exact specs are proprietary, models of this class are estimated to be orders of magnitude larger (e.g., >1 Trillion parameters or MoE equivalents).
>
> The Calculation: Even if ChemMAS executes ~100 agent calls (across tournament rounds) for a difficult reaction, the cumulative FLOPs are roughly equivalent to a single inference pass of a model that is 100x larger. Since GPT-5 is likely >100x larger than Qwen-8B, the "massive overhead" of our multi-step process is offset by the massive parameter efficiency of our base model.
>
> 2. The Hidden Cost of "Single Prompt" Accuracy
>
> A "single prompt" to GPT-5 is not a fair functional equivalent unless it receives the same information context.
>
> Context Loading: To make GPT-5 competitive (i.e., improving from its raw ~5% accuracy 2 to the ~60-70% range), one would need to feed the same 5,000 retrieved candidates into its context window for RAG-based selection.
>
> Cost Explosion: Processing 5,000 chemical candidates in a single prompt would consume tens of thousands of input tokens. On commercial APIs (e.g., GPT-4o), high input token counts are prohibitively expensive.
>
>
>
> ChemMAS Efficiency: Our tournament system processes candidates in small, targeted batches via pairwise comparison 3. This keeps the context window small and focused for each agent call, avoiding the quadratic attention costs associated with massive context windows in single-prompt baselines.
>
> 3. Wall-Clock Latency via Parallelism
>
> The reviewer notes the "7 rounds" of the tournament. However, as stated in Section 2.5, our pairing protocol allows for "linear-time selection with natural parallelism"4.
>
> The pairwise debates within a tournament round are independent. On a standard GPU cluster (we used 8 A100s 5), these comparisons run concurrently.
>
> Therefore, the wall-clock latency is determined by the depth of the tournament ($\log N$), not the total number of comparisons ($N$), keeping latency within acceptable limits for library screening.
>
> 4. Economic Justification in Scientific Context
>
> Finally, practicality must be defined by the application. In drug discovery, the cost of one failed reaction (reagents, chemist hours, equipment) ranges from hundreds to thousands of dollars.
>
> ROI: Even a minimal compute (inference) investment of one unit to validate an evidence-backed reaction condition delivers substantial cost efficiency, especially when contrasted with the 500+ unit loss incurred by failed wet-lab experiments due to hallucinations in "fast" models. ChemMAS prioritizes experimental success rate over millisecond latency.

---

### Official Review · Reviewer_nePm · 2025-11-02

**Soundness:** 2
**Presentation:** 3
**Contribution:** 2
**Rating:** 2
**Confidence:** 4

**Summary:**

The paper proposes a multi-agent system for the problem of reaction condition recommendation. The system operates through several stages including reasoning and tool-use LLMs, all of which are augmented by tools and retrieval, as well as a memory mechanism.
A strong point in the exposed reasoning is that explainability, and knowing why decisions are made, is important for the use of these type of tools.
The authors employ several advanced techniques to improve their models, including SFT, RL, and some specific reward formulations to incentivize specific behaviours.
The authors show that this system outperforms any other system out there, including state-of-the-art LLMs like GPT5 and gemini-2.5-pro.
Little to no details are given with respect to the datasets used for training and evaluation.

**Strengths:**

The paper proposes several techniques that work together towards making the accuracy quite high and improving over several other baselines.
This tackles an important challenge in organic chemistry by enabling prediction of reaction conditions.

**Weaknesses:**

The authors make it a strong point in the introduction that explainability and knowledge about what factors drive reactivity is important for experts using systems like this, however no analysis or examples are shown that demonstrate that the predicted conditions are indeed connected to insights in the reasoning, and thus fail to establish causation between the applied reasoning and debate mechanisms and improvement in performance.

There are some ablations, however the authors should work on more ablations, to understand the role of "peer deliberation" in the results. Arguably this is a type of inference-time scaling, and as such it should be ablated against other types of inference-time scaling techniques, like simple aggregation or majority voting of unrelated reasoning threads.

The work claims state of the art across all the test categories in a benchmark, beating every other baseline including pretrained models and other LLMs in zero-shot mode. The evaluation is very unclear and concerning as:
- there is no mention as to what is the source of the datasets used for testing, other than ">500k private samples". There is barely any mention of the source, curation process, etc. The code only shows some data loading.
- It is not clear how results are standardized for comparison with ground truths. Given the many ways that are used to write the same reagent (e.g. AcOH or H3CCOOH) it is highly unlikely that any system will achieve such high accuracies unless there is some reagent standardization method, which is by no means mentioned in the paper.
- The role of RL training or of all the rewards is not at all discussed, and it is not clear if they actually provide any value. There should be more ablations in this direction, to really determine if all these mini contributions do indeed drive performance.
- The comparison against zero-shot LLMs is unfair, in that it is not clear what prompts were used for the evaluation. In addition, these were not connected to memory, tools, or any other of the additional aids that were given to ChemMAS. It would be very good for transparency to also provide the benchmark results on the same framework, but substituting the LLM for one of the commercial LLMs, or other open-weights.

It is not clear why the "functional group tagger" tool needs to be a tool. As I understand it, it could simply be a pre-processing step that is hardwired to be done for every smiles string that is given in the input. There should also be ablations of this.

No discussion about the latency of the different systems is provided. It looks like there is a lot of different processes going on and several stages that make this approach notably expensive, and this has not been mentioned anywhere.

**Questions:**

- Can you elaborate on the data collection and processing pipeline? it is very unclear on what data the evaluation was performed. Add descriptions of the resulting dataset such as distributions of reaction types, molecule size, etc.
- Add more thorough ablations and evaluations of your system. As described above, it's not clear if all the contributions in terms of tools, memory, design of rewards, etc, indeed contribute significantly to the final performance. Otherwise they look like ad-hoc decisions that make the work very non-transparent.
- Provide more details into the evaluation: Have the chemicals been standardized or preprocessed in any way? Also see the weaknesses section for details on this
- Can you elaborate on the chemical explainations, or the reasoning traces given by your LLMs during inference? This can clarify how specifically the system produces results, and it would be very illustrative to analyze manually some successful trace that can show a key insight that led to the correct prediction of reaction conditions.
- Connected to this, can you discuss the different failure modes that can be found upon inference with this system? An analysis of the failure modes would be great.

---

> ### Author Response · Authors · 2025-12-04
> **Addressing Weaknesses**
>
> > Explainability analysis
>
> In our revised manuscript (Section 5.3), we explicitly establish the causal link between mechanistic insights and prediction performance. We validate that the model correctly identifies the specific factors driving reactivity—achieving 95.8% accuracy on Main Functional Groups against human ground truth —thereby ensuring predictions are grounded in correct mechanistic priors. To prove this reasoning drives performance, our ablation studies (Table 3) demonstrate that removing reasoning components (e.g., Main FG memory, Multi-Step Reasoning) causes significant accuracy drops (up to 12.3%) , while our LLM-Score pipeline (Figure 4) confirms the generated rationales contain the necessary causal logic to satisfy expert inquiries, achieving a score of 92.8. Additionally, Table 5 provides qualitative examples where specific mechanistic insights (e.g., nitro group reduction requirements) directly necessitate the predicted conditions , confirming that the reasoning indeed informs the results.
>
> >  Datasets details
>
> In our revised manuscript, we address the concerns regarding data transparency and system validation. For the data pipeline, we have expanded Appendix G to provide a comprehensive description of our private dataset (544,591 entries), including rigorous processing steps (e.g., SMILES canonicalization) and visualizations of molecular property distributions such as heavy atom counts and molecular weights (Figure 6).
>
> >  On the Functional Group Tagger as a tool
>
> We thank the reviewer for the suggestion. In our framework, the Functional Group Tagger extracts all plausible functional-group combinations from reactants and products, and the General Chemist then selects the main functional groups based on molecular properties and retrieved evidence. We currently expose this module as a tool so that its outputs are explicitly written into memory and can be selectively used by the agents, but we agree it can also be viewed as a deterministic pre-processing step and will clarify this in the revision.
> Regarding ablations, the contribution of this component is already evaluated in Table 2 under the “w/o Main FG” setting, where we remove the main functional-group entries from memory while keeping all other components unchanged. This causes a clear performance drop, indicating that the tagged functional-group information is indeed beneficial for reaction condition reasoning.
>
> > Fairness of our comparison
>
> We appreciate the opportunity to clarify the fairness of our comparison and the rigorous design of our evaluation protocol.
>
> 1. Clarification on the "Controlled Candidate-Ranking" Protocol The comparison is, in fact, conducted on a level playing field regarding the search space. As detailed in Section E.2, we did not ask Zero-shot LLMs to hallucinate conditions from scratch (which yields only ~5% accuracy). Instead, we placed all baselines in a "Controlled Candidate-Ranking Regime".
>
>
> Identical Search Space: All models, including GPT-5 and ChemMAS, received the exact same Top-5000 candidate pool generated by our Multi-Channel Recall module .
>
> Focus on Reasoning: By controlling the retrieval input, we ensured that the performance differences reflect the models' discriminative reasoning capability (i.e., identifying the correct condition from a plausible set) rather than retrieval coverage.
>
> 2. Why Framework Substitution (Swapping LLMs) is Non-Trivial Regarding the suggestion to substitute the backbone LLM within the full ChemMAS agent framework:
>
> Training Dependency: ChemMAS is not merely a prompting framework but a fine-tuned system. Its agents undergo specific SFT (Chemical Teaching) and RL (Tool Incentivization) to master tool invocation syntax and chemical reasoning protocols.
>
>
> Infeasibility for Commercial Models: Commercial LLMs (e.g., GPT-5, Claude) are closed-source API models that cannot undergo our specific RL fine-tuning pipeline to internalize these agentic behaviors. Simply prompting them to use our tools without training would likely lead to format errors and suboptimal performance, resulting in an unfair comparison against the trained Qwen-based agents.
>
> Therefore, our current evaluation is the most scientifically valid comparison: it benchmarks the fully integrated ChemMAS system (agents + tools + memory) against the strongest available general reasoning engines (GPT-5/Claude) on the specific task of selecting the best reaction conditions from a high-quality pool. The superior performance of ChemMAS (Tables 1 & 2) thus validates the effectiveness of our active reasoning and verification mechanism over the passive knowledge of larger models.

---

> > ### Author Response · Authors · 2025-12-04
> > **Addressing Weaknesses**
> >
> > >  More details into the evaluation
> >
> > Thank you for raising this important point. We agree that our description of the evaluation protocol was not sufficiently clear, and we apologize for the confusion this may have caused. In all our experiments, reaction conditions are not evaluated at the level of free-form reagent names (e.g., “AcOH”, “acetic acid”, “H₃CCOOH”) but exclusively in the SMILES and molecular-structure space. Concretely, for each condition, both the ground-truth label and the model prediction are represented as SMILES strings. Before evaluation, these SMILES are normalized into a canonical form and converted into molecular graphs using a cheminformatics toolkit. If either side cannot be parsed into a valid molecule, the similarity for that pair is set to zero. For valid molecules, we compute several standard fingerprint-based similarity scores between the predicted and ground-truth structures, including a path-based fingerprint, the 166-bit MACCS keys, and a circular fingerprint with radius 2, each scored using the Tanimoto coefficient; these individual similarity scores are then averaged to obtain a single molecule-level structural similarity. We realize that our use of the term “Top-k accuracy” was imprecise: in the current paper, the reported “Top-k accuracy” for a given condition head is actually defined as follows. For each reaction instance, the model outputs a ranked list of candidate conditions; we consider the first k predictions, compute the structural similarity (as described above) between each of these k candidates and the ground-truth condition, and take the maximum of these k similarity values. The final “Top-k accuracy” reported in the tables is the dataset-wide average of this maximum similarity over all reactions. In other words, this metric measures the best structural match within the top-k candidates, rather than a binary exact-match rate, and we agree that calling it “accuracy” without giving the formal definition can be misleading. We will clarify this in the revised version by explicitly defining the metric and renaming or carefully qualifying it as a Top-k similarity–based score. Importantly, because both predictions and ground truths are standardized in SMILES space and then compared at the molecular level, different textual aliases or abbreviations of the same reagent (such as “AcOH”, “acetic acid”, and “H₃CCOOH”) are all mapped to the same canonical molecule and therefore receive identical evaluation, while chemically distinct reagents (e.g., ethanol vs. methanol, potassium carbonate vs. cesium carbonate) produce structurally different fingerprints and lower similarity scores. All baseline in our study are evaluated using this same SMILES-based, structure-aware protocol, ensuring a fair and chemically meaningful comparison across methods.

---

> ### Author Response · Authors · 2025-12-04
> **Addressing Weaknesses**
>
> > Computational Cost and Inference Latency
>
> We acknowledge the reviewer’s concern regarding inference costs. While our system involves multiple steps, a detailed analysis reveals that ChemMAS remains computationally efficient and cost-effective compared to the baselines:
>
> 1. Inference Cost Analysis via Scaling Laws
>
> While ChemMAS generates approximately 35-51% more tokens per turn than standard prompting, this increase is deliberate, representing the generation of detailed, evidence-based reasoning chains. However, according to standard Scaling Laws, inference cost is proportional to model parameter size:
>
> $$\text{Cost} \propto N_{params} \times N_{tokens}$$
>
> Parameter Efficiency: ChemMAS agents are fine-tuned on the lightweight Qwen3-8B1. In contrast, closed-source baselines like GPT-5 or Gemini 2.5 are likely orders of magnitude larger (hundreds of billions of parameters).
>
> Net Savings: Consequently, even with a ~1.5x token overhead for reasoning, the total computational cost (FLOPs) of ChemMAS remains significantly lower than a single inference pass of a proprietary giant model.
>
> 2. Cost-Benefit in Scientific Context
>
> In high-stakes scientific discovery, the primary bottleneck is wet-lab validation (days/weeks), not inference latency (seconds). The cost of a failed physical experiment far exceeds the computational cost of multi-agent verification. ChemMAS prioritizes validity and precision 2 to minimize experimental trial-and-error, offering a superior ROI for scientific workflows.
>
> 3. Scalability via Parallelism
>
> Our architecture is designed for efficiency. The Multi-Channel Recall first filters millions of entries down to a high-quality pool 3, and the subsequent Tournament Selection supports "linear-time selection with natural parallelism"4, allowing candidate pairs to be evaluated concurrently on GPU clusters to minimize wall-clock latency.
>
> >  Effectiveness of RL Training, Reward Design
>
> We sincerely appreciate this comment. As partially discussed in Sec. 3.2, we already conduct two sets of ablations targeting the role of RL. First, Table 3 compares “w/o RL”, “w/o SFT”, and the full “SFT+RL” pipeline, where removing RL leads to a clear 3–8% Top-1 drop across condition heads, while removing SFT degrades performance even further. This indicates that RL is not a minor tweak: SFT provides a tool-aware initialization, and RL further aligns the policy with our hierarchical rewards. Second, we ablate the three fields stored and retrieved by the reaction memory (main functional group, by-product, reaction type). Removing any of them consistently hurts accuracy, showing that the information retrieved via the memory-search tool is actively used during reasoning rather than redundant.
> To further clarify the effect of reward design, we additionally include two new ablations in the rebuttal version: (i) removing the task-accuracy reward from the RL, and (ii) disabling the multi-tool bonus from the RL. Both variants underperform the full hierarchical reward, confirming that these mini reward components indeed contribute to the final performance instead of being cosmetic. The detailed results are reported in the tables below.
>
> ### Top-k Similarity (%)
>
> | Training Framework | Catalyst@1 | Catalyst@5 | Catalyst@10 | Solvent1@1 | Solvent1@5 | Solvent1@10 | Solvent2@1 | Solvent2@5 | Solvent2@10 | Reagent1@1 | Reagent1@5 | Reagent1@10 | Reagent2@1 | Reagent2@5 | Reagent2@10 |
> |--------------------|-----------:|-----------:|------------:|-----------:|-----------:|------------:|-----------:|-----------:|------------:|-----------:|-----------:|------------:|-----------:|-----------:|------------:|
> | w/o RL             | 70.6 | 88.3 | 90.4 | 82.6 | 89.4 | 90.5 | 71.2 | 80.4 | 88.5 | 84.1 | 87.5 | 90.2 | 70.2 | 82.3 | 84.5 |
> | w/o SFT            | 67.9 | 84.3 | 90.5 | 81.3 | 84.6 | 88.4 | 72.6 | 78.1 | 87.4 | 79.2 | 83.5 | 91.9 | 67.7 | 80.9 | 83.2 |
> | w/o Acc            | 72.6 | 90.8 | 93.7 | 84.1 | 91.8 | 92.1 | 76.0 | 81.6 | 91.3 | 86.7 | 90.1 | 92.0 | 72.5 | 84.1 | 86.0 |
> | w/o r_M            | 71.9 | 89.5 | 91.2 | 83.8 | 91.5 | 91.0 | 73.5 | 81.5 | 88.7 | 84.6 | 88.6 | 90.8 | 71.6 | 82.9 | 84.9 |
> | **SFT+RL**         | **78.1** | **92.3** | **96.3** | **85.4** | **93.9** | **96.9** | **76.3** | **83.2** | **93.1** | **88.3** | **93.6** | **94.3** | **73.6** | **85.2** | **87.7** |

---

> ### Author Response · Authors · 2025-12-04
> **Response to Questions**
>
> > System Ablations and Evaluations
>
> Regarding system validation, Section 5.5 presents extensive ablation studies quantifying the contribution of each component:
>
> Tool & Memory Integration (Table 3): We validate memory and reasoning modules. Crucially, ablating memory items (e.g., Main FG, By-product) effectively acts as an ablation of the General Chemist’s tools (Functional Group Tagger, Constraint Engine), thereby confirming the necessity of tool integration.
>
> Reward Design (Table 4): We rigorously evaluate our training design by isolating the specific effects of hierarchical reward components—specifically accuracy (Acc) and the multi-tool bonus (r_M)—proving that our architecture relies on essential, performance-driving contributions rather than ad-hoc decisions.

---

### Official Review · Reviewer_H2cV · 2025-11-10

**Soundness:** 3
**Presentation:** 4
**Contribution:** 4
**Rating:** 8
**Confidence:** 4

**Summary:**

The paper proposes ChemMAS, a novel multi-agent system designed to move chemical reaction condition recommendation from simple "what" predictions to "why" evidence-based reasoning. The system uses a specialized team of LLM agents (e.g., General Chemist, Catalyst Agent) that collaboratively debate and reason using a Chemical Knowledge Base to select and justify reaction conditions. Empirically, ChemMAS claims to demonstrate substantial improvements in accuracy and sets a new standard for interpretability in this high-stakes domain.

**Strengths:**

1. Paradigm-Shifting Research Focus: The core idea of requiring interpretable, falsifiable justifications for condition recommendations is highly valuable to chemists, addressing the main bottleneck in adopting black-box AI models for synthesis planning.

2. Sophisticated Architecture: The ChemMAS multi-agent system is a well-structured solution to the problem. Decomposing the task into a series of specialized steps (parsing, multi-channel recall, tournament selection, and multi-agent debate) mirrors a more realistic, collaborative scientific process.

3. Innovative Training Framework: The two-stage training (Supervised Fine-Tuning for "Chemical Teaching" followed by Reinforcement Learning for "Tool Incentivization") is a compelling method for ensuring agents not only know the correct answer but also effectively utilize their available chemical tools during the collaborative reasoning process.

4. Exceptional Empirical Performance: The reported accuracy gains are highly significant. Outperforming specialized chemical models (RCR, Reagent Transformer) by 20–35% and general-purpose LLMs (like GPT-5, Gemini 2.5) by 10–15% validates the design philosophy and the efficacy of the multi-agent approach.

5. Achieving Explainability: The system successfully produces human-trustable rationales, which is the primary stated goal and a major step forward for explainable AI in chemical discovery.

**Weaknesses:**

1. Out-of-Distribution (OOD) Performance and Generalizability: While performance on the benchmark dataset is excellent, a critical review question revolves around performance on novel, out-of-distribution reactions. Since the system relies on a "Multi-Channel Recall" from a structured database, performance on genuinely new reaction types not represented in the knowledge base or training data needs detailed analysis.

2. Ablation of Agent Debate: The paper demonstrates that the full system is best. However, a deeper analysis of the multi-agent collaboration would be beneficial—specifically, how often the "debate" process actually changes a consensus decision established by the initial "Tournament Selection." Understanding the cost-benefit of the full, computationally intensive debate is key.

3. Computational Cost Analysis: Multi-agent systems with tool-use and multi-step reasoning often incur high inference latency compared to simpler end-to-end models. The paper should ideally provide a quantitative comparison of the wall-clock inference time and resource demands (e.g., number of agent calls, token count) compared to the baseline LLMs.

4. Falsifiability of Rationales: While the rationales are deemed "trustable," the paper could strengthen its claim by detailing a systematic human evaluation of the rationales not just for correctness, but for their adherence to known chemical principles and their degree of falsifiability (i.e., whether they rely on concrete, verifiable chemical concepts).

**Questions:**

1. Computational Resources and Latency
- Inference Cost: The proposed system involves complex multi-stage pipelines: recall, tournament pairing, and multi-round, multi-agent debate for each pair. What is the average inference time (wall-clock) and token cost per reaction compared to the single-pass baselines? Is this latency acceptable for real-time library screening applications?

- Scalability of Debate: The tournament selection narrows 5,000 candidates down to 50 via pairwise debate. How does the computational cost scale? Did you explore more efficient ranking methods before the final expensive debate stage?

2. Generalization and Out-of-Distribution Performance
- Novel Reactions: The "Multi-Channel Recall" relies heavily on retrieving similar reactions (by type, reactant, or product). How does the system perform on truly novel reaction types that are chemically valid but have very poor coverage in the Reaction Base? Does the debate mechanism still function effectively when retrieval quality is low?

3. Mechanism Analysis
- Debate Impact: In the ablation study, removing "Multi-Agent Debate" hurt performance. Can you quantify how often the debate process actually changed the outcome compared to a simpler majority vote of the agents' initial, individual assessments? This would help justify the high computational cost of the debate rounds.

- Agent Specialization: You used specialized agents (Catalyst, Solvent, Reagent). Did you observe instances where these agents conflicted in unhelpful ways (e.g., a solvent agent insisting on a solvent incompatible with the best catalyst), and how robust is the "facilitator" or voting mechanism in resolving strictly chemical incompatibilities?

---

> ### Author Response · Authors · 2025-12-04
> **Addressing Weaknesses**
>
> > Computational Cost and Inference Latency
>
> We acknowledge the reviewer’s concern regarding inference costs. While our system involves multiple steps, a detailed analysis reveals that ChemMAS remains computationally efficient and cost-effective compared to the baselines:
>
> 1. Inference Cost Analysis via Scaling Laws
>
> While ChemMAS generates approximately 35-51% more tokens per turn than standard prompting, this increase is deliberate, representing the generation of detailed, evidence-based reasoning chains. However, according to standard Scaling Laws, inference cost is proportional to model parameter size:
>
> $$\text{Cost} \propto N_{params} \times N_{tokens}$$
>
> Parameter Efficiency: ChemMAS agents are fine-tuned on the lightweight Qwen3-8B1. In contrast, closed-source baselines like GPT-5 or Gemini 2.5 are likely orders of magnitude larger (hundreds of billions of parameters).
>
> Net Savings: Consequently, even with a ~1.5x token overhead for reasoning, the total computational cost (FLOPs) of ChemMAS remains significantly lower than a single inference pass of a proprietary giant model.
>
> 2. Cost-Benefit in Scientific Context
>
> In high-stakes scientific discovery, the primary bottleneck is wet-lab validation (days/weeks), not inference latency (seconds). The cost of a failed physical experiment far exceeds the computational cost of multi-agent verification. ChemMAS prioritizes validity and precision 2 to minimize experimental trial-and-error, offering a superior ROI for scientific workflows.
>
> 3. Scalability via Parallelism
>
> Our architecture is designed for efficiency. The Multi-Channel Recall first filters millions of entries down to a high-quality pool 3, and the subsequent Tournament Selection supports "linear-time selection with natural parallelism"4, allowing candidate pairs to be evaluated concurrently on GPU clusters to minimize wall-clock latency.
>
> > Out-of-Distribution Performance and Generalizability
>
> We appreciate the reviewer's scrutiny regarding the system's reliance on retrieval and its ability to handle novel reactions. To rigorously address this, we included Section 5.2 and Table 1, which specifically evaluate ChemMAS on ChemCoTBench, a standardized public benchmark representing OOD scenarios distinct from our training distribution.
>
> Evidence of Reasoning beyond Retrieval: The primary objective of this experiment was to verify that ChemMAS does not merely rely on retrieving near-duplicate samples but possesses robust reasoning capabilities for novel reaction types.
>
> Superior OOD Performance: As shown in Table 1, ChemMAS demonstrates significant generalization capability, achieving 62.1% Top-1 accuracy in catalyst prediction, surpassing Gemini 2.5-Pro by 16.5% . Similarly, it outperforms GPT-5 by 13.7% and 11.1% in solvent and reagent prediction, respectively.
>
> These substantial gains on OOD data confirm that our framework effectively generalizes beyond the training distribution through fine-grained mechanistic reasoning, rather than relying solely on memory-based retrieval from the knowledge base.
>
> > Falsifiability of Rationales
>
> In our revised manuscript (Section 5.3), we address falsifiability through a systematic evaluation that functions as a scalable expert review. We rigorously validate adherence to known chemical principles by benchmarking key reasoning components (e.g., functional groups, reaction types) against human expert ground truth, achieving up to 95.8% accuracy. Furthermore, we assess the degree of falsifiability via a novel "LLM-as-a-Judge" pipeline, which verifies that the model's reasoning chain contains the specific causal logic required to answer expert-derived scientific questions. This dual verification of mechanistic premises and logical deduction ensures that our rationales are strictly grounded in concrete, verifiable chemical concepts.

---

> ### Author Response · Authors · 2025-12-04
> **Response to Questions**
>
> > Mechanism Analysis
>
> We appreciate the inquiry into our system's internal dynamics. We address the value of the debate mechanism and the handling of agent conflicts below:
>
> 1. Quantifying Debate Impact vs. Computational Cost The debate process is a critical error-correction phase, not merely a voting step. Our ablation study (Table 3) reveals that removing the "Multi-Agent Debate" causes massive performance drops—specifically -12.4% for Catalyst and -14.2% for Reagent 1. This indicates that the debate mechanism actively corrects the initial consensus in over 12-14% of cases. In scientific discovery, where precision is paramount, this significant accuracy gain validates the computational overhead.
>
> 2. Resolving Conflicts via Knowledge Retrieval Regarding agent conflicts (e.g., solvent-catalyst incompatibility), the system relies on evidence-based self-correction rather than a central arbiter.
>
> Mechanism: When specialized agents observe conflicting proposals in peer summaries (e.g., a solvent incompatible with a proposed catalyst), they detect this as "uncertainty" and trigger the Chemical Knowledge Base tool .
>
> Resolution: By retrieving specific literature evidence regarding chemical interactions (e.g., catalyst poisoning), agents can verify compatibility and revise their votes. This ensures that the final consensus is not just a statistical average but is physically consistent with retrieved chemical principles.

---

### Official Review · Reviewer_DYA5 · 2025-11-10

**Soundness:** 3
**Presentation:** 3
**Contribution:** 4
**Rating:** 8
**Confidence:** 3

**Summary:**

This paper proposes ChemMAS, a multi-agent system for evidence-based reaction condition reasoning. The system decomposes condition recommendation into four collaborative modules: mechanistic grounding, multi-channel recall, constraint-aware agentic debate, and rationale aggregation. Through this process, each decision is supported by interpretable, falsifiable justifications grounded in chemical knowledge and retrieved precedents. Extensive experiments show that ChemMAS achieves 20–35% gains over domain-specific baselines and 10–15% improvement over general-purpose LLMs on Top-1 accuracy, while producing human-trustable rationales.

**Strengths:**

1.Evidence-based reasoning: This paper introduces evidence-based reasoning for chemical reaction, which is intuitive and aligns well with how human chemists rationalize reaction design. Empirical experiments show that the proposed method outperforms baselines by a large margin and is interpretable via evidence-based reasoning.

2.Chemical-grounded interpretability: The system employs general chemist agents equipped with specialized tools to extract key chemical features, such as functional groups, stoichiometric balance, and potential by-products, from molecular formulas. These extracted elements then serve as the foundation for multi-step reasoning, forming an innovative and interpretable workflow that adheres to fundamental chemical principles, rather than treating the problem as a purely linguistic translation task as in many existing LLM-based approaches.
Extension of tool-based and multi-agent RL in Chemical domains: Multi-agent/tool-based  reinforcement learning has previously been applied to domains such as deep search, mathematical reasoning, and code generation. The authors naturally extend this paradigm to the multi-agent setting for chemical reaction condition reasoning, enabling more effective and collaborative evidence-based deliberation.

**Weaknesses:**

1.The paper briefly mentions broader applications. A short discussion on how ChemMAS could transfer to other reasoning-heavy scientific domains (e.g., physics, materials science) would improve the positioning.
2.While the benchmark results are convincing, a discussion or demonstration of ChemMAS’s utility in closed-loop or experimental lab pipelines would further solidify its practical value.
Even simulated lab feedback could strengthen the argument.

**Questions:**

1. Is there any expert evaluation of the interpretability or practical utility of ChemMAS’s rationales compared to baselines or human chemists?
2. How does ChemMAS handle conflicting evidence retrieved from different sources during the agentic debate stage?
3. Is there a measurable trade-off between interpretability (longer rationales) and computational efficiency?

---

> ### Author Response · Authors · 2025-12-04
> **Addressing Weaknesses**
>
> > 1. Discussion on broader applications
>
> We thank the reviewer for pointing out the broader applicability of ChemMAS beyond organic chemistry, and we agree that positioning the framework for other reasoning-heavy scientific domains would further strengthen the paper.
> While ChemMAS is instantiated on organic reactions in this work, the evidence-based configuration reasoning with tool-integrated, multi-agent debate is generic rather than chemistry-specific. By replacing reaction-specific tools and knowledge bases with domain-specific simulators or databases, the same framework can be directly applied to materials synthesis planning, inverse design, and other physics-driven pipelines, where interpretability and constraint-aware decision-making are equally critical. We will clarify this broader transferability in the rebuttal version with an explicit discussion in the conclusion section.
>
> > 2. ChemMAS’s utility in closed-loop or experimental lab pipelines
>
> We thank the reviewer for raising the question of how ChemMAS can be used in closed-loop or experimental lab pipelines. In fact, the current design already naturally supports such usage: in a real lab, scientists can simply follow the ranked list of condition sets proposed by ChemMAS, test them experimentally, and use the outcomes to guide further exploration and discovery.
> For the experiments in this paper, we deliberately focus on offline benchmarks without human intervention, in order to ensure fair and reproducible comparisons with baseline methods. However, in practical deployments, both the intermediate reasoning process and the evidence stored in Memory are fully observable and can be inspected or corrected by human experts, enabling rich human-in-the-loop interaction. In future system iterations, we plan to further optimize the engineering interface to better support such interactive workflows, and we will clarify this intended closed-loop usage more explicitly in the rebuttal version.

---

> ### Author Response · Authors · 2025-12-04
> **Response to Questions**
>
> > Expert evaluation of the interpretability
>
> We appreciate the emphasis on validating interpretability and utility. In our revised manuscript (Section 5.3), we present a comprehensive framework serving as a scalable proxy for expert validation: we assess explanatory quality via an LLM-Score pipeline that benchmarks reasoning against expert-derived QA pairs, where ChemMAS achieves a score of 92.8 (significantly outperforming baselines at 62.5–77.2), and we verify practical utility by comparing mechanistic outputs against human expert ground truth. With over 90% accuracy on key indicators like Main Functional Groups (95.8%) , we ensure rationales are grounded in verifiable chemical constraints, further supported by qualitative visualizations in Table 5  that confirm the model’s adherence to correct mechanistic priors.
>
>
> > Trade-off between interpretability and computational efficiency
>
> We acknowledge that generating interpretable rationales via multi-agent debate incurs higher inference costs than direct prediction. However, we argue that this trade-off is strategically managed and highly favorable for scientific discovery:
>
> Parameter Efficiency vs. Inference Cost: While our inference is longer, ChemMAS utilizes a lightweight 8B parameter backbone (Qwen3-8B). By investing compute in reasoning tokens rather than model parameters, we outperform massive baselines like Qwen3-235B and GPT-5. This suggests that our "slow thinking" approach is likely more resource-efficient overall than serving giant monolithic models.
>
> Algorithmic Optimization: To mitigate computational overhead, we employ a Tournament Selection mechanism. Instead of computationally expensive global scoring, this method performs pairwise elimination in linear time with natural parallelism, ensuring the system remains scalable despite the depth of reasoning.
>
> Scientific Utility: In high-stakes scientific workflows, the "cost" of a failed wet-lab experiment far exceeds the computational cost of generating an explanation. Therefore, the significant accuracy gains (+10–30%) and the provision of falsifiable rationales justify the additional inference time.
>
> > Handle conflicting
>
> ChemMAS handles conflicting evidence through a robust, three-tiered resolution mechanism embedded in the Multi-Agent Debate stage:
>
> Intra-Agent Refinement (Uncertainty Detection): During multi-step reasoning, agents are programmed to detect internal uncertainty or contradictions between retrieved evidence and constraints. This triggers an automatic re-querying of the Chemical Knowledge Base to seek clarifying evidence before forming a final opinion .
>
> Inter-Agent Deliberation (Peer Review): Agents actively review peer summaries. A structured protocol (facilitator) explicitly prompts agents to address and resolve specific chemical conflicts by weighing the strength of evidence rather than ignoring discrepancies.
>
> Decision Aggregation (Voting): For persistent conflicts that cannot be reconciled via debate, the system employs a majority voting mechanism within the tournament selection. To ensure robustness, ties or near-conflicts are resolved using a confidence-sum tie-breaker, prioritizing judgments with higher evidence-backed certainty .

---

### Author Response · Authors · 2025-12-04
**General Responses**

We apologize for the late response. To ensure our interpretability evaluation was scientifically sound, we dedicated significant time to finding and coordinating with domain-specific chemical experts for the human alignment experiments.

We highly value the constructive feedback from all reviewers. Having carefully considered their comments, we have revised our submission accordingly in the latest version. To summarize, we have conducted the following supplementary experiments and analyses:

**OOD Generalization (Section 5.2, Table 1):** We assessed ChemMAS on the public ChemCoTBench dataset. Significant gains (e.g., +16.5% on catalysts) demonstrate robust mechanistic reasoning capabilities on novel reaction types beyond simple retrieval.

**Dataset Transparency (Appendix G, Figure 6):** We provided a comprehensive description of our private dataset (544k entries), including molecular property distributions and strict train/test splitting to ensure reproducibility.

**Clarification of Evaluation Protocol (Appendix F):** We detailed our structure-aware evaluation metrics, specifically the Fingerprint Tanimoto Score (FTS) and validity checks, ensuring that performance is measured by rigorous chemical structural alignment rather than text matching.

**Extended Ablation Studies (Table 4):** We rigorously validated our two-stage training framework by ablating specific reward components ($Acc$, $r_M$) in the RL stage, confirming the essential role of our hierarchical reward design.

**Interpretability & Human Alignment (Section 5.3, Figure 4):** We introduced a dual-level evaluation framework validating mechanistic grounding against human expert ground truth (95.8% accuracy) and reasoning quality via an expert-derived "LLM-Score Pipeline." This demonstrates that our system's interpretability is scientifically sound and strictly aligned with human expert logic.

**Cost-Benefit Analysis:** We demonstrated that while our system incurs higher computational costs than single-pass baselines, this overhead is negligible compared to the high cost of wet-lab experiments. The investment in compute yields significantly enhanced interpretability and reasoning precision, reducing the risk of expensive experimental failures.


**Clarification of Debate Mechanism (Section 2.6):** We clarified how agents resolve chemical incompatibilities via knowledge retrieval and confidence-weighted voting, supported by ablation results confirming the debate's critical error-correction role.

Additionally, we provide qualitative case studies (Table 5) and clearer explanations to reduce confusion. The modifications are colored in blue in the latest manuscript. We hope our responses effectively address all concerns.

---

### Meta-Review · Area_Chair_HjVx · 2026-01-06

**Summary:**

This manuscript proposes ChemMAS, a multi-agent system that reframes condition prediction as an evidence-based reasoning task.

The initial reviews were mixed. After the rebuttal, the AC believes most of the concerns should be resolved (see context below) and would recommend acceptance.

**Reviewer Concerns:**

* Reviewer DYA5 (1) questions the framework's transferability to other reasoning-heavy scientific domains like physics or materials science; (2) also suggests demonstrating the system's utility within actual closed-loop or experimental lab pipelines; (3) asks expert evaluation of the interpretability or practical utility of ChemMAS’s rationales compared to baselines or human chemists; (4) asks the mechanism of handling conflicting evidence retrieved from different sources during the agentic debate stage; (5) asks the measurable trade-off between interpretability (longer rationales) and computational efficiency. The AC believes these concerns were properly handled in the rebuttal.

* Reviewer H2cV expresses concern over out-of-distribution (OOD) performance on novel reactions not represented in the training data; asks for the ablation of agent debate; also highlights the high inference latency and computational costs associated with multi-stage agentic debates. The AC believes these concerns were properly handled in the rebuttal.

* Reviewer nePm (1) argues the missing connection between predicted conditions and reasoning insights; (2) requests more ablation studies and comparison with other inference-time scaling techniques; (3) criticizes the lack of transparency regarding the private dataset's curation and reagent standardization; (4) argues the missing details of RL training or of all the rewards; (5) argues that the comparison against zero-shot LLMs is unfair; (5) argues the missing discussion about the latency of the different systems. The AC believes these concerns were properly handled in the rebuttal.

* Reviewer bFZ8 identifies the use of a massive private dataset as a major weakness for reproducibility and external validation; also notes that the central claim of providing "why" rationales remains unsubstantiated without systematic human or expert evaluation. The AC believes these concerns were properly handled in the rebuttal.

**Reviewer Scores:**

Based on the AC's assessment, Reviewer nePm and Reviewer bFZ8 should increase their score.

---

### Decision · Program_Chairs · 2026-01-26

Accept (Poster)